# SSFL: Discovering Sparse Unified Subnetworks at Initialization for Efficient Federated Learning

**Riyasat Ohib**[†][*]                                                              *riyasat.ohib@gatech.edu*
*Georgia Institute of Technology*

**Bishal Thapaliya**[†][*]                                                          *bthapaliya16@gmail.com*
*TReNDS Center*

**Gintare Karolina Dziugaite**                                                     *gkdz@google.com*
*Google DeepMind, Mila - Quebec AI Institute*

**Jingyu Liu**[†]                                                                  *jliu75@gsu.edu*
*Georgia State University*

**Vince D. Calhoun**[†]                                                            *vcalhoun@gatech.edu*
*Georgia State University, Georgia Institute of Technology, Emory University*

**Sergey Plis**[†]                                                                 *s.m.plis@gmail.com*
*Georgia State University*

[†]*Center for Translational Research in Neuroimaging & Data Science (TReNDS), Georgia State University, Georgia Institute of Technology, and Emory University.*

**Reviewed on OpenReview:** *https://openreview.net/forum?id=kUZ6LhUB26*

## Abstract

In this work, we propose Salient Sparse Federated Learning (SSFL), a streamlined approach for sparse federated learning with efficient communication. SSFL identifies a sparse subnetwork prior to training, leveraging parameter saliency scores computed separately on local client data in non-IID scenarios, and then aggregated, to determine a global mask. Only the sparse model weights are trained and communicated each round between the clients and the server. On standard benchmarks including CIFAR-10, CIFAR-100, and Tiny-ImageNet, SSFL consistently improves the accuracy–sparsity trade-off, achieving more than 20% relative error reduction on CIFAR-10 compared to the strongest sparse baseline, while reducing communication costs by 2× relative to dense FL. Finally, in a real-world federated learning deployment, SSFL delivers over 2.3× faster communication time, underscoring its practical efficiency. Our code is available at: https://github.com/riohib/SSFL

**Keywords:** Federated Learning, Sparse Neural Networks, Model Pruning, Pruning at Initialization, Sparse Subspaces, Communication Efficiency, Non-IID Data.

## 1 Introduction

The success of deep learning has been propelled by ever-larger models trained on centralized datasets. Yet, in critical domains such as healthcare, finance, and mobile computing, data is inherently decentralized, privacy-sensitive, and distributed across millions of user devices or institutional silos. Federated Learning (FL)

---

[*]These authors share first authorship. Correspondence to: `riyasat.ohib@gatech.edu`; `bthapaliya16@gmail.com`.

has emerged as a compelling paradigm for such settings, enabling collaborative model training without direct data sharing (McMahan & Ramage, 2017). By orchestrating learning across clients who retain their local data, FL promises privacy preservation and data locality. However, practical deployment of FL, especially in cross-device scenarios, remains fraught with challenges.

FL systems must contend with severe *system heterogeneity*, where client devices exhibit widely varying compute, memory, and energy resources. This is compounded by *statistical heterogeneity*, as data across clients is typically non-IID and differently balanced, making it difficult to train a single global model that generalizes well across users (Kairouz et al., 2021). These challenges often manifest as communication bottlenecks, poor convergence behavior, and brittle hyperparameter tuning (Khodak et al., 2021).

One promising approach to alleviating these bottlenecks is through sparse training, which reduces both the on-device computation and the communication payload (Lin et al., 2017; Wang et al., 2020; Evci et al., 2020a). Among sparse FL methods, there has been considerable progress in dynamic sparsification strategies that evolve sparse masks over the course of training (Bibikar et al., 2022; Dai et al., 2022; Guastella et al., 2025). However, these methods typically rely on iterative coordination, complex heuristics such as prune-and-grow schedules, and introduce additional hyperparameters that require tuning. Moreover, certain pruning-at-initialization (PaI) approaches resort to using public proxy datasets to identify subnetworks, which violates the privacy-preserving ethos of FL (Huang et al., 2022). Beyond operational complexity, dynamic masking approaches may suffer from a more fundamental challenge: as masks evolve independently across clients, parameter updates occur in shifting subspaces, potentially hindering the formation of coherent global representations.

In this work, we address a fundamental open problem in sparse and efficient FL: how can one identify a performant, static sparse subnetwork at initialization using only private, non-IID data across clients with minimal communication and no auxiliary dataset? To this end, we propose Salient Sparse Federated Learning (SSFL), a simple yet effective method that discovers a globally shared sparse mask prior to training via a single step of distributed saliency aggregation at the start of the learning process. Each client computes local gradient-based parameter importance scores on private data and transmits them *once* to the server, which averages them in a data-proportional manner to construct a global heatmap of saliency scores. The Top-$k$ operation on the aggregated saliency heatmap defines a fixed binary mask, establishing a common sparse subnetwork for all clients that is preserved for the duration of training. By constraining the federation to this unified sparse topology, we ensure optimization proceeds within a consistent parameter subspace, avoiding the potential instability of shifting masks found in dynamic approaches. This leads to a highly communication-efficient and privacy-compliant static sparse FL pipeline, with zero overhead beyond the initial mask discovery.

Extensive experiments across CIFAR-10, CIFAR-100, and Tiny-ImageNet under realistic non-IID settings demonstrate that SSFL significantly outperforms both dense and sparse FL baselines, including state-of-the-art dynamic sparsity methods such as DisPFL (Dai et al., 2022), SparsyFed (Guastella et al., 2025) and Flash (Babakniya et al., 2023). Moreover, we validate the practical efficiency of SSFL in a real-world FL deployment, achieving over 2.3× wall-clock communication speedups on larger models. To the best our knowledge, SSFL is the first method to offer a single-shot, fully decentralized, and privacy-preserving sparse FL strategy with competitive performance across standard benchmarks.

**Contributions** We summarize our contributions as follows:

- We introduce SSFL, the first single-shot sparse federated learning framework that discovers a globally shared sparse subnetwork at initialization using only local client data. Our approach avoids the need for public auxiliary datasets, iterative pruning cycles and additional hyperparameters.

- We develop a communication-efficient mechanism to compute globally representative importance scores by aggregating local saliency scores in a single round of coordination. This aggregation respects both data heterogeneity and client privacy, providing a practical solution for stable sparsity in federated learning.

- We demonstrate that SSFL consistently outperforms state-of-the-art sparse federated learning baselines, including recent dynamic methods such as DisPFL and SparsyFed (Section 4.1). On CIFAR-10, SSFL achieves over 20% relative error reduction compared to the strongest sparse baseline, and it delivers similarly strong accuracy–sparsity improvements across CIFAR-100, Tiny-ImageNet, and diverse non-IID partitions. On ResNet-50, the performance gap widens over DisPFL from +23% at 50% sparsity to +35% at 95% sparsity, where DisPFL collapses to 12% while SSFL maintains around 48% accuracy.

- We validate SSFL's effectiveness through ablation studies showing: ① SSFL approaches oracle salient subspace quality with increasing client participation in Section 4.2.3 ② the discovered subspaces encode meaningful structure, as evidenced by performance degradation under permutation in Section 4.2.2 ③ the framework can adapt to out-of-distribution data when needed in Section 4.3.

- We deploy SSFL in a real-world federated learning system and observe over 2× faster wall-clock communication compared to dense baselines (Section 4.1.1), underscoring the method's practical impact.

## 2 Related Work

Our work lies at the intersection of model pruning and communication-efficient federated learning (FL). We group related efforts based on how they impose sparsity and highlight how SSFL enables one-shot global sparse training using only private, non-IID data and a single round of communication.

**Dynamic sparsity in federated learning.** A growing body of work explores dynamic sparsity, where sparse masks evolve during training. These methods are designed to adapt to changing data distributions but typically require repeated coordination between clients and the server. For example, SparsyFed (Guastella et al., 2025) combines activation pruning with adaptive reparameterization, while DisPFL (Dai et al., 2022) and FedDST (Bibikar et al., 2022) adopt the RigL strategy (Evci et al., 2020a), periodically regrowing pruned weights during training. DSFL (Beitollahi et al., 2022) introduces user-adaptive compression rates, MADS (Yan et al., 2025) proposes mobility-aware sparsification that adjusts based on contact time and model staleness, and DSFedCon (Li et al., 2025) integrates dynamic sparse training with federated contrastive learning.

Although effective, such methods introduce additional hyperparameters and require multiple rounds of communication for mask updates. Moreover, because clients maintain different and evolving masks, the server must aggregate updates defined over partially non-overlapping parameter subspaces, complicating aggregation and often leading to denser effective global models. In contrast, SSFL discovers a single global mask at initialization, ensuring that all updates are aligned in the same sparse subspace and avoiding iterative coordination altogether. This also provides a more stable optimization trajectory, as clients remain in a common subspace throughout training.

**Structured sparsity and specialization.** Some FL methods impose structured sparsity by pruning filters, blocks, or layers to reduce memory and inference costs on constrained devices. HeteroFL (Diao et al., 2021) assigns clients submodels based on compute budgets, while AdaptCL (Zhou et al., 2021) employs adaptive structured pruning to synchronize FL processes across heterogeneous environments. Sub-FedAvg (Vahidian et al., 2021) combines both structured and unstructured pruning for personalized federated learning under data heterogeneity. More recent work includes subMFL (Oz et al., 2024), which generates compatible submodels through server-side structured pruning, and AdaPruneFL (Fan et al., 2024), which proposes data-free adaptive structured pruning for heterogeneous client capabilities.

While structured sparsity primarily targets computational efficiency, the dominant bottleneck in federated learning lies in communication. Unstructured sparsity better addresses this by maximizing parameter reduction without hardware-specific constraints. Importantly, the sparse local models produced by SSFL can still benefit from unstructured-sparsity-aware accelerators (e.g., SPMM kernels), enabling device-level speedups(Nakahara et al., 2019; Thangarasa et al., 2023; Gale et al., 2020; NVIDIA, 2021; 2020a) . Thus, whereas structured approaches emphasize client specialization and hardware compatibility, SSFL focuses on global communication savings while remaining compatible with hardware acceleration.

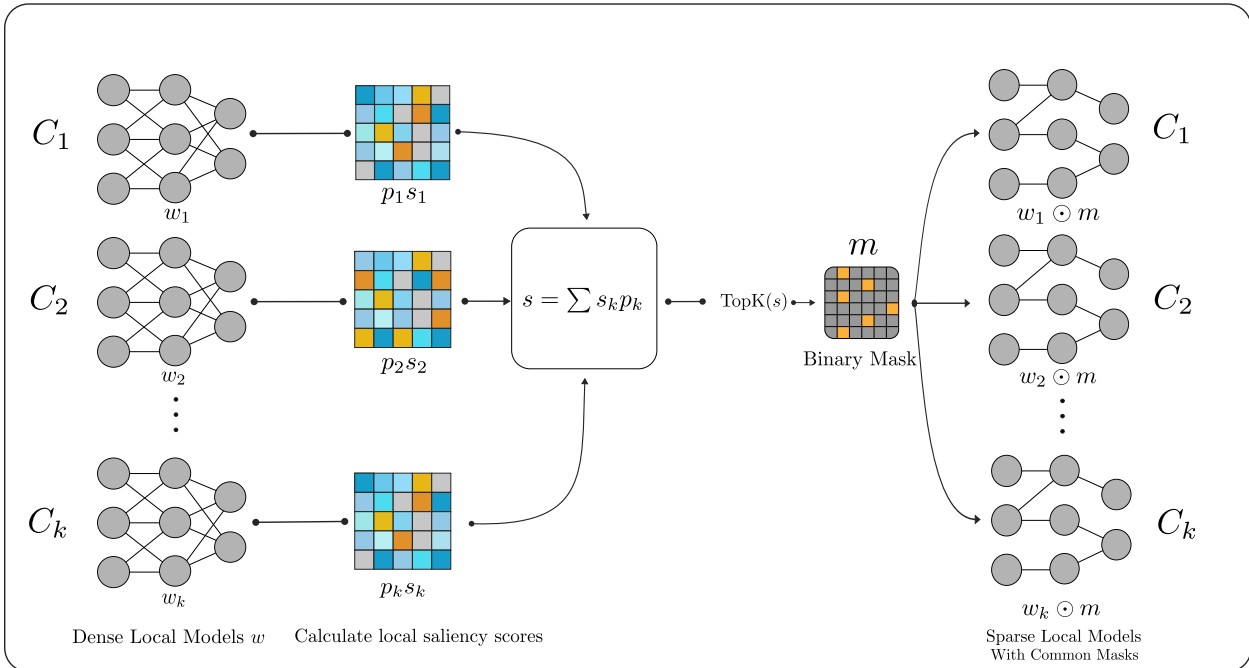

Figure 1: Illustration of the distributed connection importance in the non-IID setting. The parameter saliency scores from each site calculated on local minibatches of equal class distribution are aggregated, weighing them with the proportion of the data available at that site. The common mask generated from that score is applied to local client models.

**Static sparsity and pruning-at-initialization in FL.** Pruning-at-initialization (PaI) techniques aim to identify a sparse subnetwork before training begins. Although well-studied in centralized training (Lee et al., 2018; Wang et al., 2020; Tanaka et al., 2020), their application in federated learning remains limited. Several methods circumvent the challenge by relying on public proxy datasets to estimate parameter importance, as in FedTiny (Huang et al., 2022). Others compute masks locally but struggle to generalize across clients under non-IID distributions (Jiang et al., 2022). FedPaI (Wang et al., 2025) proposes a progressive, multi-round pruning strategy, but this introduces additional pruning stages and communication costs. SSFL addresses this gap by introducing a single-shot, privacy-preserving approach to identify a high-quality global mask using only local client data. To the best of our knowledge, it is the first to demonstrate that gradient-based saliency scores, when aggregated keeping the data disparity in mind, can enable static sparse training in FL without auxiliary public datasets or iterative coordination during the training process.

A more detailed survey of related pruning and sparse federated learning approaches, including additional variants and historical context, is provided in Appendix B.

## 3 The Salient Sparse Federated Learning (SSFL) Method

Federated learning poses unique challenges for sparsity: decentralized data, non-IID samples, and limited communication make it difficult to identify globally effective sparse subnetworks. In particular, pruning-at-initialization (PaI) strategies, while effective in centralized settings, require careful adaptation to function under these constraints. SSFL addresses this challenge by introducing a single-shot, communication-efficient method that constructs a shared sparse subnetwork before training begins. The approach requires only a single round of client–server interaction, using local saliency scores to discover a globally important subset of parameters. Once the mask is generated, all further training proceeds on the same sparse subnetwork using standard federated averaging.

Our method consists of two main phases: (i) a one-time sparse mask discovery phase based on distributed gradient saliency aggregation, and (ii) a sparse training phase where clients perform local updates and global aggregation restricted to the space of parameters selected through the mask.

### 3.1 Parameter Saliency at Initialization

We begin by defining the importance criterion used to identify salient parameters prior to training. Saliency-based pruning methods estimate how much each parameter contributes to the loss at initialization. This idea, initially proposed in early neural network pruning work (Mozer & Smolensky, 1988) and refined in methods like SNIP (Lee et al., 2018), is central to SSFL's mask discovery phase.

Given a randomly initialized model $\boldsymbol{w}_0 \in \mathbb{R}^d$ and a loss function $\mathcal{L}(\boldsymbol{w})$ evaluated on a dataset $\mathcal{D}$, the saliency score $s_j$ for a parameter $w_j$ is defined as:

$$s_j = \left| \frac{\partial \mathcal{L}(\boldsymbol{w}_0; \mathcal{D})}{\partial w_j} \cdot w_j \right|. \tag{1}$$

This score captures the first-order sensitivity of the loss to removing parameter $w_j$ at initialization. Parameters with larger magnitudes of $s_j$ are considered more important and are prioritized for retention.

In SSFL, each client computes this saliency score locally on a minibatch sampled from its private data. Because this estimation is done only once, at initialization, the method incurs no additional runtime overhead during training. The main challenge lies in extending this centralized scoring mechanism to a federated setting in a way that respects data heterogeneity and avoids repeated communication.

### 3.2 The SSFL Process: Single-Shot Mask Discovery and Sparse Training

SSFL extends the saliency criterion from centralized training to federated settings through a single-shot distributed process. This involves computing local importance scores at each client, aggregating them in a data-weighted manner to produce a global importance vector, and selecting the top-scoring parameters to define a shared sparse subnetwork. The full procedure consists of four stages:

**Local Saliency Estimation**

At initialization, all clients synchronize to a common model $\boldsymbol{w}_0 \in \mathbb{R}^d$. Each client $k$ samples a minibatch $\mathcal{B}_k \subset \mathcal{D}_k$ of size $B$ from its private data and computes the local empirical loss for the obtained batch

$$\mathcal{L}(\boldsymbol{w}_0; \mathcal{B}_k) = \frac{1}{B} \sum_{(\boldsymbol{x}, y) \in \mathcal{B}_k} \ell(\boldsymbol{w}_0; \boldsymbol{x}, y). \tag{2}$$

Using this loss, client $k$ computes local saliency scores $\boldsymbol{s}_k \in \mathbb{R}^d$ by applying the importance criterion in Equation (1) to its minibatch:

$$s_{k,j} = \left| \frac{\partial \mathcal{L}(\boldsymbol{w}_0; \mathcal{B}_k)}{\partial w_{0,j}} \cdot w_{0,j} \right|.$$

This process is fully local and requires no communication. We use only a single minibatch per client to minimize computation and preserve privacy. A detailed analysis of the saliency estimation is provided in Appendix A.1.

**Aggregation of Saliency Scores.**

Each client transmits its saliency scores $\boldsymbol{s}_k$ and local dataset size $n_k = |\mathcal{D}_k|$ to the server. A global saliency vector $\boldsymbol{s} \in \mathbb{R}^d$ is then computed via a weighted average:

$$\boldsymbol{s} = \sum_{k=1}^{K} p_k \boldsymbol{s}_k, \quad \text{where} \quad p_k = \frac{n_k}{\sum_{i=1}^{K} n_i}. \tag{3}$$

This data-weighted aggregation ensures that clients with more examples exert proportionally greater influence on the final importance scores, which helps mitigate heterogeneity bias. The class-balanced minibatch sampling further ensures that the computed saliency scores are representative of all classes available in the local clients.

### Global Mask Selection.

Given a target sparsity level $\sigma \in (0, 1)$, the number of active parameters is $\hbar = \lfloor (1 - \sigma) \cdot d \rfloor$. A binary mask $\mathbf{m} \in \{0, 1\}^d$ is then computed as:

$$\mathbf{m} = \text{TopK}(\boldsymbol{s}, \hbar),$$

where $\text{TopK}(\cdot, \hbar)$ returns a binary vector with ones at the indices of the $\hbar$ largest values in $\boldsymbol{s}$. This mask is broadcast to all clients and remains fixed throughout training.

### Sparse Federated Training.

Training proceeds in standard rounds of local updates and model aggregation, with all operations restricted to the masked subnetwork. At communication round $r$, each participating client $k$ receives the global sparse model $\boldsymbol{w}^r_{g,m}$ and performs $T$ local update steps.

During training, both forward and backward propagation operate exclusively on the sparse subnetwork defined by the mask $\mathbf{m}$. At local step $t$, the forward pass computes the loss using only the active (non-zero) parameters:

$$\mathcal{L}(\boldsymbol{w}^t_{k,m} \odot \mathbf{m}; \mathcal{B}_{k,t}),$$

where $\boldsymbol{w}^t_{k,m} \odot \mathbf{m}$ ensures that masked-out parameters contribute zero to the network computation. The corresponding masked gradient update is then applied as:

$$\boldsymbol{w}^{t+1}_{k,m} \leftarrow \boldsymbol{w}^t_{k,m} - \eta \left( \nabla_{\boldsymbol{w}} \mathcal{L}(\boldsymbol{w}^t_{k,m} \odot \mathbf{m}; \mathcal{B}_{k,t}) \odot \mathbf{m} \right),$$

where $\mathcal{B}_{k,t}$ is a local minibatch. This ensures that gradients are computed and applied only for the active parameters, while masked parameters remain fixed at zero throughout training.

After $T$ steps, each client sends its sparse model back to the server, and aggregation proceeds as in standard FedAvg. Because the mask is fixed, all communication and computation can be compressed accordingly using standard sparse encodings (e.g., CSR, COO, or bitmask formats); we discuss the communication accounting protocol in detail in the Section 4.

### 3.3 Dataset and non-IID partition

We evaluate SSFL on CIFAR-10, CIFAR-100 Krizhevsky et al. (2009) and the TinyImagenet dataset. For simulating non-identical data distributions across the federating clients we use two separate data partition strategies. We next present these non-IID data partition generating strategies and provide a more detailed treatment in Appendix A.2.

**Dirichlet Partition** We use the Dirichlet distribution to simulate non-IID distribution of data among the clients Hsu et al. (2019) and call this the *Dirichlet Partition* in our experiments. We assume that each client selects training examples independently, with class labels distributed across $N$ classes according to a categorical distribution defined by the vector $\boldsymbol{q}$, where each $q_i \geq 0$ for $i \in \{1, 2, 3, ..., N\}$ and $\|\boldsymbol{q}\|_1 = 1$.

To model a diverse set of clients, each with distinct data, we draw $\boldsymbol{q} \sim \text{Dir}(\alpha \boldsymbol{p})$ from the Dirichlet distribution to determine the class distribution vector $\boldsymbol{q}$. Here, the vector $p$ establishes the baseline distribution across the $N$ classes, while the *concentration* parameter $\alpha > 0$, dictates the degree of variation or the property of IID in class distribution among clients. By adjusting $\alpha$, we can simulate client populations with varying degrees of data uniformity. Specifically, as $\alpha \to \infty$, the class distributions of the clients converge to the baseline prior distribution $p$; conversely, as $\alpha$ is chosen to be smaller, increasing fewer classes chosen at random dominates the proportion of data at each class, representing a higher degree of non-IID distribution. As $\alpha \to 0$, each client holds data from one particular class chosen at random.

---

**Algorithm 1** SSFL: One-shot sparse subnetwork discovery and training

---

**Require:** Number of clients $K$, communication rounds $R$, local update steps $T$, sparsity level $\sigma$
**Ensure:** Final sparse global model $\boldsymbol{w}_{g,m}^R$
    **Phase 1: One-Time Mask Generation**
  1: Server initializes model $\boldsymbol{w}_0$ and broadcasts it to all clients.
  2: **for all** clients $k \in \{1, \ldots, K\}$ **in parallel do**
  3:      Sample a minibatch $\mathcal{B}_k \subset \mathcal{D}_k$
  4:      Compute local saliency vector $\boldsymbol{s}_k$ using equation 1
  5:      Send $\boldsymbol{s}_k$ and local data size $n_k = |\mathcal{D}_k|$ to the server
  6: **end for**
  7: Server computes data proportions $p_k = n_k / \sum_{i=1}^K n_i$
  8: Server aggregates global saliency: $\boldsymbol{s} \leftarrow \sum_{k=1}^K p_k \boldsymbol{s}_k$
  9: Server sets number of active parameters $\Bbbk \leftarrow \lfloor (1 - \sigma) \cdot d \rfloor$
10: Server generates global mask $\mathbf{m} \leftarrow \text{TopK}(\boldsymbol{s}, \Bbbk)$
11: Server broadcasts mask $\mathbf{m}$ to all clients
12: Server initializes sparse global model: $\boldsymbol{w}_{g,m}^0 \leftarrow \boldsymbol{w}_0 \odot \mathbf{m}$
    **Phase 2: Sparse Federated Training**
13: **for** $r = 0$ to $R - 1$ **do**
14:      Server selects a subset of clients $\mathcal{C}' \subseteq \{1, \ldots, K\}$
15:      Server sends current sparse global model $\boldsymbol{w}_{g,m}^r$ to all clients in $\mathcal{C}'$
16:      **for all** clients $k \in \mathcal{C}'$ **in parallel do**
17:         Initialize local model $\boldsymbol{w}_k \leftarrow \boldsymbol{w}_{g,m}^r$
18:         **for** $t = 0$ to $T - 1$ **do**
19:            Sample minibatch $\mathcal{B}_{k,t} \subset \mathcal{D}_k$
20:            Compute masked gradient: $\nabla \mathcal{L}(\boldsymbol{w}_k; \mathcal{B}_{k,t}) \odot \mathbf{m}$
21:            Update local model: $\boldsymbol{w}_k \leftarrow \boldsymbol{w}_k - \eta \nabla \mathcal{L} \odot \mathbf{m}$
22:         **end for**
23:         Send updated local model $\boldsymbol{w}_k$ to server
24:      **end for**Server aggregates client models: $w_{g,m}^{r+1} \leftarrow \sum_{k \in C'} \frac{n_k}{\sum_{j \in C'} n_j} w_k$
25: **end for**

---

In this work, we partition the training data according to a Dirichlet distribution $\text{Dir}(\alpha)$ for each client and generate the corresponding test data for each client following the same distribution. We set the prior distribution $p$ to be uniform over $N$ classes and specify the $\alpha = 0.3$ for CIFAR10 and $\alpha = 0.2$ for the CIFAR-100 for fair comparison to previous work (Dai et al., 2022; Bibikar et al., 2022). A detailed description of the Dirichlet partitioning scheme with a figure depicting client data allocation is provided in Appendix A.2.

**Pathological Partition** Following prior work on non-IID federated learning settings (McMahan et al., 2017; Zhang et al., 2020), we simulate pathological data partitions, where each client is randomly assigned only a limited number of classes from the total number of classes. In particular, each client receives at most 2 classes for CIFAR-10 and 10 classes for CIFAR-100.

## 4 Experiments

We evaluate SSFL against a broad range of federated learning methods to demonstrate its effectiveness. Our experimental design is four-fold: (1) We first compare SSFL with state-of-the-art sparse and dense FL baselines on non-IID distributions of CIFAR-10, CIFAR-100, and TinyImageNet. (2) We then study the performance of SSFL under varying sparsity levels up to 95% and compare it to other sparse FL methods. (3) We deploy our method in a real-world FL framework to report wall-clock time improvements. (4) Finally, we analyze the structural properties of the discovered sparse mask by studying the effect of intra-layer permutations and the quality of the generated mask.

## 4.1 Main Results

Our primary results are organized by the data partitioning strategy to clearly delineate performance under different non-IID conditions. All experiments in this subsection use a ResNet18 architecture with 50% sparsity.

### 4.1.1 Performance on varying dataset and partitioning schemes

**Performance on Dirichlet Partition.** Table 1 shows the performance of all methods under the Dirichlet non-IID data partition. On CIFAR-10, SSFL achieves the highest accuracy (88.29%) among all sparse methods and also outperforms the strongest dense baseline, FedAvg-FT (88.02%). On CIFAR-100, SSFL reaches 61.37% accuracy, again surpassing all sparse and dense techniques. This corresponds to more than a 20% relative error reduction compared to the strongest sparse baseline (DisPFL), underscoring the substantial margin by which SSFL improves over prior work. These results demonstrate that the global mask discovered by SSFL effectively captures salient features that generalize well across clients with heterogeneous data distributions. Results are averaged over five runs.

**Performance on Pathological Partition.** We further stress-test our method using a pathological non-IID partition, where each client holds data from a very limited number of classes. The results, shown in Table 2, highlight SSFL's robustness. SSFL achieves the highest accuracy on CIFAR-10 with 94.61%. On CIFAR-100, its performance (52.01%) is highly competitive with the top-performing dense baseline, FedAvg-FT (52.47%), while significantly exceeding other sparse methods like DisPFL (44.74%). This indicates that SSFL's saliency aggregation mechanism can construct a high-quality global subnetwork even when clients have extremely skewed, non-overlapping data distributions.

**Performance on TinyImageNet.** To evaluate scalability, we conducted experiments on the more challenging Tiny-ImageNet dataset. As detailed in Table 2, SSFL again achieves the best performance with 19.4% accuracy, outperforming both dense and sparse counterparts. For this dataset we report results for the strongest baselines from our CIFAR-10/100 experiments. These result confirm that SSFL's data-driven subnetwork discovery is effective not just on smaller datasets but also scales successfully to more complex image classification tasks.

**Experiments on the larger ResNet-50 architecture** To demonstrate the scalability of SSFL beyond standard benchmarks, we conduct experiments using ResNet-50 on CIFAR-100. This represents a significantly more challenging optimization landscape with a deeper architecture. We compared SSFL against the dense baseline (FedAvg) and the strongest sparse baseline (DisPFL) across the full sparsity spectrum ($50\% \rightarrow 95\%$).

Table 1: Comprehensive comparison of SSFL with FL and sparse FL baselines on CIFAR-10 and CIFAR-100 under a **Dirichlet partition** using ResNet18 at 50% sparsity. Highlighted values indicate the best performance.

| Method | CIFAR-10 (%) | CIFAR-100 (%) | Comms (MB) | Sparse |
|---|---|---|---|---|
| FedAvg | 86.04 ±1.35 | 59.30 ±2.02 | 446.9 | ✗ |
| FedAvg-FT | 88.02 ±0.57 | 59.42 ±3.72 | 446.9 | ✗ |
| D-PSGD-FT | 83.05 ±1.99 | 50.27 ±1.61 | 446.9 | ✗ |
| Ditto | 83.50 ±0.99 | 43.50 ±2.04 | 446.9 | ✗ |
| FOMO | 66.21 ±1.54 | 32.50 ±3.65 | 446.9 | ✗ |
| **SSFL** | **88.29 ±0.42** | **61.37 ±2.01** | 223.4 | ✓ |
| DisPFL | 85.12 ±1.05 | 59.21 ±1.97 | 224.0 | ✓ |
| SparsyFed | 80.94 ±1.20 | 50.64 ±2.10 | 224.0 | ✓ |
| Flash | 81.13 ±0.95 | 51.81 ±2.35 | 224.0 | ✓ |
| SubFedAvg | 76.50 ±1.74 | 47.25 ±2.84 | 346.6 | ✓ |
| Random | 41.61 ±1.62 | 48.62 ±1.33 | 223.4 | ✓ |
| Fed-PM | 46.44 ±2.50 | 15.88 ±3.10 | optimal | optimal |

Table 2: Performance comparison on Pathological partition (CIFAR-10/100) and Dirichlet partition (TinyImageNet) using ResNet18 at 50% sparsity. Highlighted values indicate best performance.

| Pathological Partition | | | | | TinyImageNet (Dirichlet) | | | |
|---|---|---|---|---|---|---|---|---|
| Method | CIFAR-10 (%) | CIFAR-100 (%) | Comms (MB) | Sparse | Method | Acc. (%) | Comms (MB) | Sparse |
| FedAvg | 92.48 ±0.25 | **52.47 ±2.68** | 446.9 | ✗ | FedAvg | 16.92 ±0.43 | 446.9 | ✗ |
| FedAvg-FT | 92.90 ±1.33 | **52.47 ±2.04** | 446.9 | ✗ | FedAvg-FT | 18.21 ±0.48 | 446.9 | ✗ |
| D-PSGD-FT | 88.74 ±1.60 | 27.58 ±1.88 | 446.9 | ✗ | D-PSGD-FT | 11.98 ±0.62 | 446.9 | ✗ |
| Ditto | 83.54 ±1.28 | 51.90 ±2.20 | 446.9 | ✗ | Ditto | 17.80 ±0.39 | 446.9 | ✗ |
| FOMO | 88.25 ±1.10 | 45.25 ±2.09 | 446.9 | ✗ | FOMO | 4.27* | 446.9 | ✗ |
| **SSFL** | **94.61 ±1.00** | 52.01 ±1.64 | 223.4 | ✓ | **SSFL** | **19.4 ±0.30** | 223.4 | ✓ |
| DisPFL | 91.25 ±1.10 | 44.74 ±1.56 | 224.0 | ✓ | DisPFL | 8.27 ±0.29 | 224.0 | ✓ |
| SubFedAvg | 91.20 ±0.60 | 46.04 ±2.89 | 346.6 | ✓ | SubFedAvg | 18.76 ±0.28 | 346.6 | ✓ |
| Random | 55.34 ±1.25 | 46.22 ±4.59 | 223.4 | ✓ | Random | 16.76 ±0.53 | 223.4 | ✓ |

*Single run; method failed to converge.

The results in Table 3 demonstrate that SSFL's advantages amplify with the increase in model size when compared DisPFL, the best performing dynamic method in our analysis so far. SSFL consistently outperforms DisPFL across all sparsity levels, with the performance gap widening as networks become sparser from a +24.39% improvement at 50% sparsity to +35.49% at 95% sparsity. Notably, at 70% sparsity, SSFL achieves 60.82% accuracy, coming within 1.21% of the dense baseline while reducing communication by 3.3×. At extreme sparsity (95%), DisPFL's dynamic masking collapsed to 12.04%, while SSFL maintains good performance at 47.53%. These results indicate that a stable, globally shared sparse subspace becomes increasingly critical as models scale and sparsity increases.

Table 3: ResNet-50 on CIFAR-100. SSFL significantly outperforms DisPFL, with the performance gap widening at higher sparsity levels.

| Method | 50% | 70% | 80% | 90% | 95% |
|---|---|---|---|---|---|
| **FedAvg (Dense)** | 62.03% | 62.03% | 62.03% | 62.03% | 62.03% |
| **DisPFL** | 36.55% | 32.96% | 30.65% | 21.87% | 12.04% |
| **SSFL (Ours)** | **59.76%** | **60.82%** | **56.73%** | **55.93%** | **47.53%** |
| Improvement | +23.21% | +27.86% | +26.08% | +34.06% | +35.49% |

**Experiment Details and Baselines.** We use the SGD optimizer for all techniques with a weight decay of 0.0005. All methods use 5 local epochs, except for Ditto, which uses 3 epochs for the local model and 2 for the global model. The initial learning rate is 0.1, decaying by a factor of 0.998 after each communication round. We use a batch size of 16 for all experiments. A total of $R = 500$ global communication rounds are executed for CIFAR-10 and CIFAR-100.Following standard practice (Dai et al., 2022; Liu et al., 2025; Lin et al., 2018), we report communication cost under the values-only encoding assumption, where only non-zero weights are transmitted each round. This convention is conservative for SSFL, which broadcasts its global mask only once, while dynamic methods must repeatedly transmit mask indices, thereby incurring additional communication costs. We nevertheless adopt values-only encoding for all methods to ensure fair and widely comparable benchmarks, with detailed overhead analysis provided in Appendix A.4.

Our dense baselines include FedAvg (McMahan et al., 2017), FedAvg-FT (Cheng et al., 2021), Ditto (Li et al., 2021), FOMO(Zhang et al., 2020), and D-PSGD (Lian et al., 2017). Our sparse baselines are Dis-PFL(Dai et al., 2022), SubFedAvg (Vahidian et al., 2021), SparsyFed (Liu et al., 2025), Flash (Babakniya et al., 2023) and FedPM (Isik et al., 2022). Among the sparse baselines, FedPM (Isik et al., 2022) is unique in that it freezes the randomly initialized dense network and instead trains a *probability mask* over connections. The final subnetwork and its sparsity emerge from this process rather than being preset, so we report its communication and sparsity as *optimal* in Table 1. A detailed explanation of the baselines and experimental setup is available in Appendix A.8.

**Effect of varying levels of sparsity on performance**   We evaluate how SSFL's performance varies across sparsity levels from 50% to 95% on CIFAR-10 and CIFAR-100, as shown in Figure 2. We compare SSFL against sparse FL baselines DisPFL, SubFedAvg, and FedPM, along with a random masking baseline. SSFL consistently outperforms all sparse baselines across the entire sparsity spectrum. Notably, it maintains robust performance even at high sparsity levels where other methods degrade significantly, indicating that the global mask captures informative and transferable structure. Unlike DisPFL, which aggregates heterogeneous masks into a denser global model, SSFL's shared mask both simplifies training and preserves a truly sparse global model that can further benefit from hardware acceleration (e.g., SPMM kernels on GPUs/TPUs and custom accelerators) (Nakahara et al., 2019; Thangarasa et al., 2023; Gale et al., 2020; NVIDIA, 2021; 2020a).

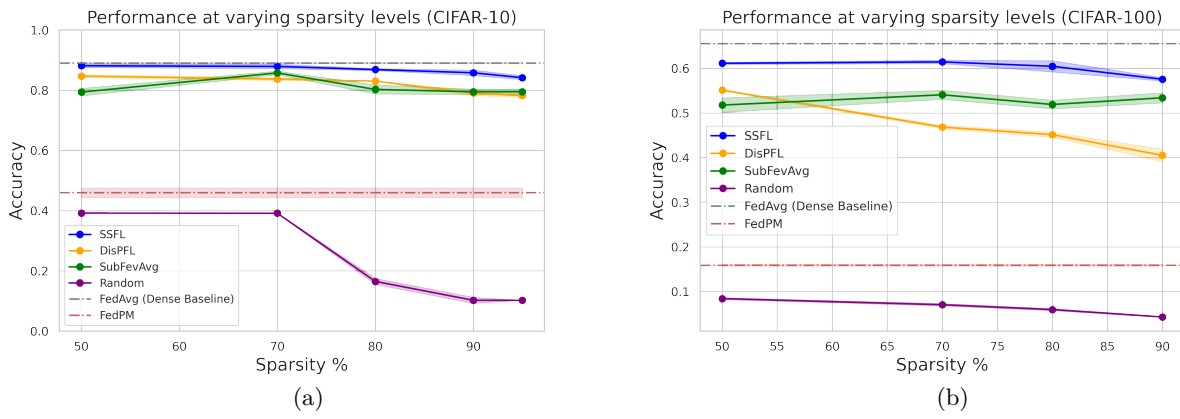

Figure 2: Performance comparison at varying levels of sparsity for SSFL with similar sparse FL methods on ResNet18 on the (a) CIFAR-10 and (b) CIFAR-100 non-IID dataset.

**Wall-clock gains under real-world network conditions.**   We deployed experiments on Amazon Web Services (AWS) across five geographically distributed regions (North Virginia, Ohio, Oregon, London, and Frankfurt). This setup emulates a realistic FL environment where clients experience substantial network latencies. We measured the average server aggregation time per round, i.e., the time required to collect all client model updates. As shown in Figure 3 (b), SSFL consistently reduces communication time relative to dense communication (FedAvg), with the gap widening as model size increases. At 90% sparsity, SSFL achieves a speedup of approximately $2.3\times$ on the 1.73M-parameter model, reducing the average aggregation latency from 1.91 s to 0.82 s per round. This highlights its practical impact in real-world deployments.

## 4.2   Structural Alignment and Robustness of the Global Sparse Mask

A central tenet of SSFL is the enforcement of a single, globally shared sparse structure discovered at initialization using limited data. We investigate the validity of this design choice by examining: ① the necessity of structural alignment compared to local masking ② the importance of specific mask topology versus layer-wise density ③ the sufficiency of our single-minibatch estimation strategy and ④ the mask's generalization across heterogeneous clients.

### 4.2.1   Global vs. Local Masks: A Case for Structural Alignment

A natural alternative to SSFL is to allow each client to maintain its own sparse subnetwork. Indeed, using client-specific masks is intuitively appealing and has been a popular choice in prior dynamic sparse training works (Dai et al., 2022; Liu et al., 2025; Evci et al., 2020a).

To isolate the importance of structural consensus, we conducted a controlled experiment comparing a *shared global random mask* against *unique client-wise random masks* (see Figure 3-a). The results demonstrate an interesting outcome: the global random mask significantly outperforms the client-specific local random masks. This highlights an important insight: *structural alignment within a unified subspace is essential for*

*effective aggregation.* By enforcing a shared global mask, SSFL ensures that all clients operate within the same sparse subspace. This enables SGD to optimize a consistent set of parameters from start to finish, avoiding the optimization instability and destructive interference caused by the haphazard aggregation of disjoint parameter updates from misaligned subspaces, which occurs in dynamic masking approaches.

### 4.2.2 Importance of mask topology: Effect of intra-layer mask permutation

While the previous experiment confirms the effectiveness of a shared global structure, it raises a secondary question: does the *specific topology* of the mask matter, or merely the sparsity ratios allocated to each layer? Previous work in centralized pruning (Frankle et al., 2021) suggests that for pruning-at-initialization, identifying correct layer-wise densities is often the primary driver of performance. We test this in the non-IID FL setting by randomly shuffling the SSFL mask within each layer (preserving layer-wise density but destroying specific structural topology). As shown in Figure 3 (a), the shuffled variant outperforms the uniform global random mask, confirming that our aggregated saliency scores successfully identify optimal layer-wise capacities. Importantly, however, the original, unshuffled SSFL mask consistently performs best, indicating that SSFL captures meaningful information about individual connection importance beyond just the layer-wise sparsity ratio.

### 4.2.3 Global Saliency Estimation via $K$ Distributed Minibatches

SSFL requires only one local minibatch per client to construct the global sparse mask, but this does not mean the mask is computed from a single minibatch overall. With $K$ participating clients, the server aggregates saliency scores from $K$ *distinct minibatches*, which in our cross-device setting (e.g., $K = 100$) corresponds to several thousand samples, comparable to the data budgets used in single-shot pruning methods such as SNIP. Since saliency is derived from first-order gradients, which concentrate rapidly with sample size, aggregating gradients across clients produces a stable and low-variance estimate of the global saliency distribution, even under severe non-i.i.d. partitions.

**Empirical validation.** We further quantify this by approximating an oracle mask $M^\star$ computed using the full training set, and evaluating how well masks estimated from $k \in \{1, \ldots, 400\}$ minibatches recover $M^\star$. As shown in Figure 4 (a-b), the mask error decays sharply and plateaus after roughly 80–100 minibatches on both CIFAR-10 and CIFAR-100, with additional data yielding negligible improvements beyond this point. In our experiments, following the standard setting in the literature (Dai et al., 2022), we use $K = 100$ clients, meaning SSFL aggregates exactly $K = 100$ minibatches (one per client) and thus operates precisely in this convergence regime. This provides strong evidence that one minibatch per client is sufficient to recover a near-oracle sparse subnetwork at initialization when the client pool is sufficiently large ($K \gtrsim 80$), enabling

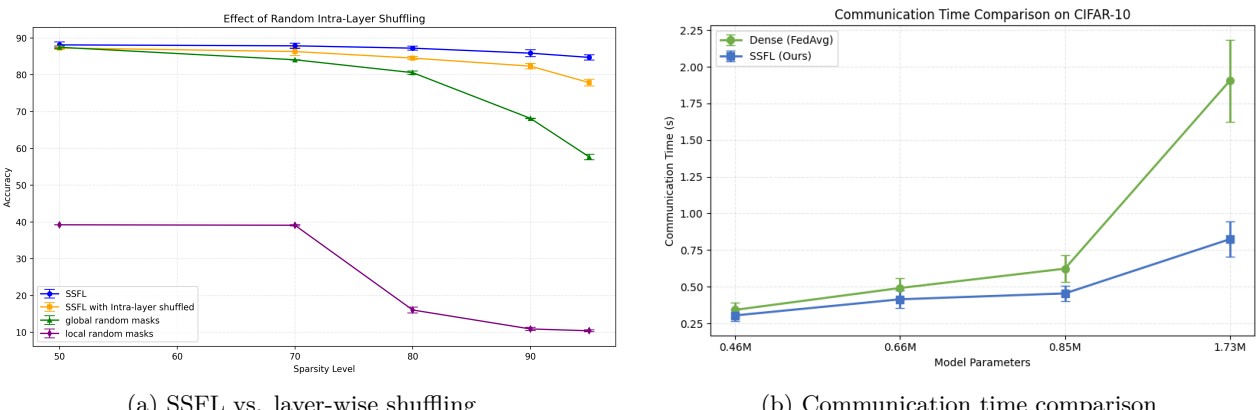

(a) SSFL vs. layer-wise shuffling                (b) Communication time comparison

Figure 3: Analysis of SSFL in different scenarios. (a) Effect of random intra-layer shuffling on SSFL masks. (b) Wall-time communication comparison between baseline dense communication and SSFL on CIFAR-10 across ResNet models of varying complexities.

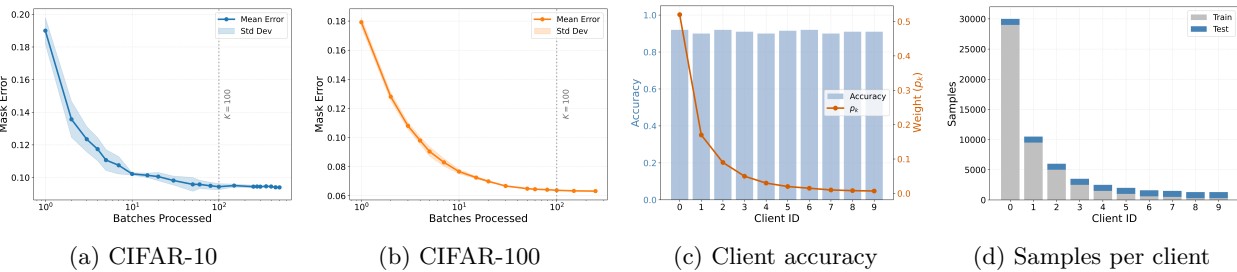

|           |            |                  |                    |
|-----------|------------|------------------|--------------------|
| (a) CIFAR-10 | (b) CIFAR-100 | (c) Client accuracy | (d) Samples per client |

Figure 4: (a-b) **Mask Convergence.** Mask error relative to the oracle decreases exponentially, stabilizing near $K \approx 80$–$100$ (vertical line). We use $K = 100$ for all experiments. (c-d). Experiments done with 5 random seeds. **Client Performance.** Local accuracy (c, bars) plotted against aggregation weights $p_k$ (orange line), alongside sample counts (d). SSFL maintains high performance even on minimal data partitions (e.g., Clients 8 and 9).

SSFL to achieve high accuracy without any iterative mask refinement or additional communication. The framework naturally extends to settings with fewer clients by increasing the number of minibatches per client to maintain oracle mask approximation quality.

### 4.2.4 Quality of the discovered mask in the non-IID setting.

We examine the effect of a shared global mask on local model performance under uneven client data distributions. Figure 4 (c-d) shows that, despite large disparities in client data volumes, SSFL yields competitive accuracy even for clients with very limited data. This indicates that aggregated saliency scores capture structure that generalizes across heterogeneous clients rather than overfitting to high-data participants. We further confirm this in Appendix A.10, which reports local accuracy alongside class distributions for random clients as well as the top and bottom-performing sites.

### 4.3 Out-of-Distribution (OOD) Adaptation

The one-time mask computation in SSFL facilitates adaptation to non-stationary environments, such as shifts in data distribution or the arrival of new clients with novel classes. To handle these scenarios, we extend SSFL to *One-Shot OOD Adaptation*, a mechanism that triggers a mask refresh upon detecting distribution shifts. To identify the new optimal subnetwork, clients compute saliency scores on the full local model state by temporarily unmasking pruned parameters, which retain their original initialization values. This approach allows the model to recruit connections outside the set of current mask if necessary for incoming data with negligible overhead compared to fully dynamic methods.

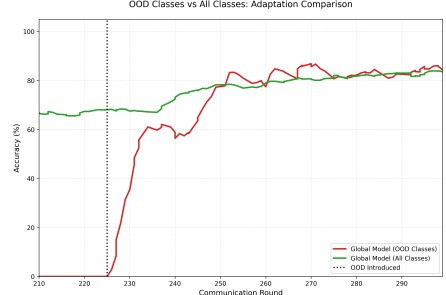

Figure 5: OOD classes are introduced at round 225 (dotted vertical line). Following the single mask update, the model rapidly acquires the new concepts (Red curve), rising from 0% to over 80% accuracy, while maintaining stable global performance (Green curve).

Figure 5 validates this approach where two new OOD classes (classe 8 and 9) are introduced at round 225. Following the single mask update, the global model rapidly adapts to the new classes (Red curve), rising from 0% to over 80% accuracy, while maintaining stable global performance (Green curve). This alludes to SSFL accommodating to distribution shifts through a single, efficient topology update. Further analysis on OOD adaptation, including the OOD Algorithm 2 is detailed in Appendix A.6 including direction of future work.

### 4.4 Further Analysis

We provide further analysis of SSFL's design choices, computational analysis, sensitivity to warmup and practical deployment factors in Appendix A. Specifically, we detail computational complexity in Appendix A.3

and communication encoding costs in Appendix A.4, demonstrating SSFL's theoretical and empirical efficiency gains over dynamic methods. Additionally, Appendix A.7 investigates the sensitivity of saliency estimation to initialization by comparing our standard random initialization against warmup strategies.

## 5 Limitations and Future Work

While SSFL demonstrates the effectiveness of static sparse subnetwork discovery, several natural extensions remain. First, while we introduced a one-shot OOD adaptation mechanism to handle distinct distribution shifts, the core SSFL framework prioritizes stability via a static mask. In scenarios with highly volatile, continuous concept drift, more granular or automated adaptation strategies may be required to balance the trade-off between mask stability and plasticity.

Second, our warmup analysis (Appendix A.7) validates the robustness of initialization-based mask discovery, showing negligible performance differences ($\pm 1\%$) compared to post-warmup approaches. This finding reinforces the stability of our method, consistent with the low variance observed across experiments with five different random seeds. However, investigating alternative saliency criteria or ensembles beyond our gradient-based approach may yield even stronger subnetworks in distributed non-IID settings.

Beyond these limitations, we identify several promising research directions. While our experiments on ResNet-50 demonstrate that SSFL's advantages over dynamic methods amplify with model scale (widening from $+23\%$ to $+35\%$ as sparsity increases), extending this to foundation models remains an exciting avenue. Furthermore, extending SSFL to structured sparsity patterns could unlock greater speedups on modern accelerators. SSFL naturally complements differential privacy, since it requires only a single noisy saliency aggregation step, thereby minimizing privacy breach while preserving communication efficiency. Finally, integrating personalization on top of a shared global mask may provide more robust performance under extreme heterogeneity across clients.

## 6 Conclusion

We presented Salient Sparse Federated Learning (SSFL), a streamlined framework that mitigates communication bottlenecks in federated learning by identifying a high-quality global sparse subnetwork before training. SSFL requires only a single, privacy-preserving round of communication where local gradient-based saliency scores are aggregated into a shared mask. This static initialization eliminates the hyperparameter tuning and iterative coordination typical of dynamic sparsity methods.

SSFL achieves substantial communication savings across CIFAR-10, CIFAR-100, and Tiny-ImageNet, while matching or exceeding dense baselines. More importantly, we demonstrated that these benefits scale to deeper architectures (ResNet-50), where SSFL maintains superior accuracy-sparsity trade-offs compared to dynamic approache DisPFL. Furthermore, our introduction of a one-shot OOD adaptation mechanism shows that SSFL has the flexibility to readily handle distribution shifts without abandoning the efficiency of a static mask during training.

Real-world deployment on geographically distributed AWS regions further shows up to $2.3\times$ wall-clock speedups, underscoring practical efficiency. These results challenge the assumption that complex, iterative mask adjustments are required for strong sparse FL performance. Instead, they demonstrate that a well-chosen static subnetwork, identified collectively at initialization, enables a simpler, more stable, and highly effective training trajectory. Looking ahead, SSFL offers a robust foundation for hybrid static–adaptive refinements, personalized sparse models, and structured sparsity for hardware acceleration.

## 7 Broader Impact

This paper presents work whose goal is to advance the field of Machine Learning. Our primary objective is to study and improve efficiency associated with distributed federated learning.

**Broader Impact Statement**

This work aims to improve the efficiency of federated learning by reducing communication and computation costs. More efficient FL can broaden access to collaborative training in settings with limited resources, such as mobile or healthcare applications, and may help reduce the environmental footprint of large-scale distributed training. As with any federated approach, care is required to ensure privacy protections (e.g., differential privacy or secure aggregation) and to prevent efficiency gains from coming at the expense of marginalized users.

**Acknowledgments**

We thank the action editor and the reviewers for their insightful comments that helped us improve the paper.

This work was supported by NIH R01DA040487 and in part by NSF 2112455, and NIH 2R01EB006841

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

# A  Appendix

## A.1  Detailed analysis of the Connection Saliency Criterion

This section provides a detailed derivation of the gradient-based connection saliency metric used in SSFL, as defined in Equation (1). The fundamental goal of pruning at initialization is to identify the most important parameters in a randomly initialized network, $\boldsymbol{w}_0$, without performing any training. This line of work traces back to the early idea of skeletonization by Mozer and Smolensky (Mozer & Smolensky, 1988) and was later refined in single-shot pruning methods such as SNIP (Lee et al., 2018).

The importance of a single parameter, $w_{0,j}$, can be defined as the change in the total loss, $\mathcal{L}$, that would occur if that parameter were removed from the network. Formally, this change $\Delta\mathcal{L}_j$ is:

$$\Delta\mathcal{L}_j = \mathcal{L}(\boldsymbol{w}_0; \mathcal{D}) - \mathcal{L}(\boldsymbol{w}_0 - w_{0,j}\boldsymbol{e}_j; \mathcal{D}), \tag{4}$$

where $\boldsymbol{e}_j$ is the one-hot vector for the $j$-th parameter. However, computing this value for every parameter is computationally prohibitive, as it would require $d+1$ forward passes over the data $\mathcal{D}$, where $d$ is the number of parameters.

To create a tractable importance score, we approximate $\Delta\mathcal{L}_j$ using a first-order Taylor expansion. This approach reframes the question from "what is the effect of removing a parameter?" to "how sensitive is the loss to the presence of a parameter at initialization?". To formalize this, we introduce an auxiliary "gating" vector $\mathbf{c} \in [0,1]^d$, which multiplies the model's weights $\boldsymbol{w}_0$. The loss is now a function of the masked weights: $\mathcal{L}(\mathbf{c} \odot \boldsymbol{w}_0; \mathcal{D})$.

The sensitivity of the loss to the $j$-th parameter can be measured by the derivative of $\mathcal{L}$ with respect to $c_j$, evaluated at $\mathbf{c} = \mathbf{1}$ (the initial state where all connections are fully active). This derivative, which we denote as the saliency $g_j$, approximates the effect of a small perturbation on connection $j$. We can compute it directly using the chain rule:

$$g_j(\boldsymbol{w}_0; \mathcal{D}) = \left.\frac{\partial\mathcal{L}(\mathbf{c} \odot \boldsymbol{w}_0; \mathcal{D})}{\partial c_j}\right|_{\mathbf{c}=\mathbf{1}} \tag{5}$$

$$= \left.\frac{\partial\mathcal{L}(\boldsymbol{w}_0; \mathcal{D})}{\partial w_{0,j}} \cdot \frac{\partial(c_j w_{0,j})}{\partial c_j}\right|_{\mathbf{c}=\mathbf{1}} \tag{6}$$

$$= \frac{\partial\mathcal{L}(\boldsymbol{w}_0; \mathcal{D})}{\partial w_{0,j}} \cdot w_{0,j}. \tag{7}$$

This result, $g_j$, provides a computationally efficient, single-pass estimate of the importance of parameter $w_{0,j}$. The final saliency score $s_j$ used in SSFL is the magnitude of this term, preserving parameters that have the largest estimated impact on the loss:

$$s_j = |g_j(\boldsymbol{w}_0; \mathcal{D})| = \left|\frac{\partial\mathcal{L}(\boldsymbol{w}_0; \mathcal{D})}{\partial w_{0,j}} \cdot w_{0,j}\right|. \tag{8}$$

It is important to note that while these scores could be normalized (e.g., by dividing by their sum), this step is unnecessary for our method, as the Top-k selection procedure only depends on the relative ranking of the scores.

## A.2  Generating non-IID data partition with Dirichlet Distribution

In this section, we provide the necessary background on generating non-identical data distribution in the client sites using the Dirichlet Distribution, specifically for the context of federated learning.

**non-IID data in FL**  Federated Learning (FL), as introduced by McMahan & Ramage (2017), is a framework designed for training models on decentralized data while preserving privacy. It utilizes the Federated Averaging (FedAvg) algorithm where each device, or client, receives a model from a central server, performs stochastic gradient descent (SGD) on its local data, and sends the models back for aggregation.

Unlike data-center training where data batches are often IID (independent and identically distributed), FL typically deals with non-IID data distributions across different clients. Hence, to evaluate federated learning it is crucial to not make the IID assumption and instead generate non-IID data among clients for evaluation Hsu et al. (2019).

**Generating non-IID data from Dirichlet Distribution**  In this study, we assume that each client independently chooses training samples. These samples are classified into $N$ distinct classes, with the distribution of class labels governed by a probability vector $q$, which is non-negative and whose components sum to 1, that is, $q_i > 0$, $i \in [1, N]$ and $\|q\|_1 = 1$. For generating a group of non-identical clients, $q \sim \mathrm{Dir}(\alpha p)$ is drawn from the Dirichlet Distribution, with $p$ characterizing a prior distribution over the $N$ classes and $\alpha$ controls the degree of identicality among the existing clients and is known as the *concentration parameter*.

In this section, we generate a range of client data partitions from the Dirichlet distribution with a range of values for the concentration parameter $\alpha$ for exposition. In Figure 6, we generate a group of 10 balanced clients, each holding equal number of total samples. Similar to Hsu et al. (2019) the prior distribution $p$ is assumed to be uniform across all classes. For each client, given a concentration parameter $\alpha$, we sample a $q$ from $\mathrm{Dir}(\alpha)$ and allocate the corresponding fraction of samples from each client to that client. Figure 6 illustrates the effect of the concentration parameter $\alpha$ on the class distribution drawn from the Dirichlet distribution on different clients, for the CIFAR-10 dataset. When $\alpha \to \infty$, identical class distribution is assigned to each classes. With decreasing $\alpha$, more non-identicalness is introduced in the class distribution among the client population. At the other extreme with $\alpha \to 0$, each class only consists of one particular class.

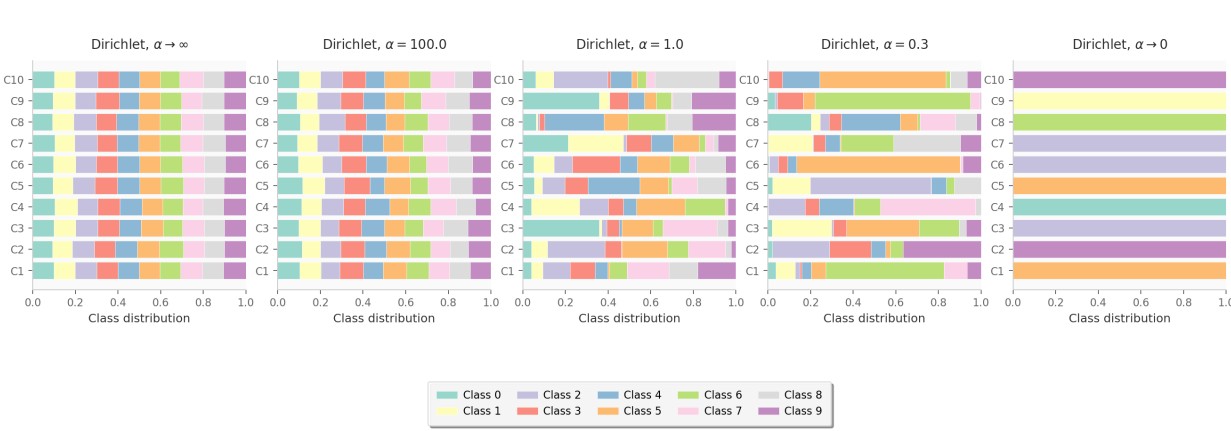

Figure 6: Generating non-identical client data partitions using the Dirichlet Distribution for the CIFAR-10 dataset among 10 clients. Distribution among classes is represented using different colors. (a) Dirichlet, $\alpha \to \infty$ results in identical clients (b–d) Client distributions generated from Dirichlet distributions with different concentration parameters $\alpha$ (e) Dirichlet, $\alpha \to 0.0$ results in each client being assigned only one class.

### A.3 Computational Complexity and Communication Analysis

To rigorously evaluate the efficiency of SSFL, we analyze the computational cost under two distinct execution regimes: *Standard Dense Execution* and *Accelerated Sparse Execution*. Let $P$ denote the number of model parameters, $E$ the number of local epochs, $B$ the number of batches per epoch, and $s \in (0, 1]$ the density of the model. We decompose the total cost per training round into three components: gradient computation ($\mathcal{C}_{\mathrm{grad}}$), dynamic topology overhead ($\mathcal{C}_{\mathrm{overhead}}$), and communication cost.

Dynamic sparse training methods incur structural overheads to update masks, such as sorting, pruning, and regrowth that often bottleneck accelerated execution. SparsyFed incurs a high overhead, as it applies

weight re-parameterization and activation pruning during every forward pass of every batch, scaling as $\mathcal{C}_{\text{overhead}} \approx O(EB \cdot P)$. Iterative pruning methods like DisPFL and the FLASH-JMWST variant rely on periodic prune-and-regrow cycles that require magnitude-based sorting of weights, resulting in a moderate overhead of $\mathcal{C}_{\text{overhead}} \approx O(P \log P)$ per round. In contrast, SSFL (Ours) and FLASH-SPDST utilize a static mask. Consequently, the topology remains fixed during the training phase, eliminating dynamic overhead entirely ($\mathcal{C}_{\text{overhead}} = 0$).

A critical distinction lies in the initialization cost. While FLASH-SPDST shares the per-round efficiency of a static mask, it requires a computationally expensive "Warm-up" Stage involving $E_d$ epochs of training on a subset of clients to estimate sensitivity. Conversely, SSFL is single-shot, estimating saliency using only one minibatch per client at initialization (requiring $K \sim 80 - 100$ clients for mask convergence, a constant factor). Since $1 \ll E_d$, SSFL is significantly more lightweight to deploy.

In terms of communication bandwidth, SSFL transmits only the non-zero values ($sP$) in the uplink since the indices are fixed and globally synchronized. Dynamic methods such as DisPFL, FLASH-JMWST, and SparsyFed lack this global index synchronization, requiring the transmission of both values and updated mask indices ($sP + \text{Idx}$). Uniquely, because SSFL and FLASH-SPDST enforce a fixed global mask, the server broadcasts only the updated sparse values ($sP$), achieving symmetric sparsity in the downlink. In contrast, DisPFL and FLASH-JMWST typically require synchronizing dense or dynamic structures, increasing the downlink load to $O(P)$ or requiring index transmission.

Table 4: Per-round computational complexity and communication costs. SSFL achieves the minimum accelerated compute cost and communication volume without the dynamic overheads of SparsyFed/JMWST or the initialization warm-up cost of FLASH-SPDST.

| Method | Accelerated Compute | Dynamic Overhead | Init. Cost | Communication | |
|---|---|---|---|---|---|
| | | | | *Uplink* | *Downlink* |
| **FedAvg** | $O(1 \cdot P)$ | $-$ | None | $O(P)$ | $O(P)$ |
| **SSFL** | $O(s \cdot P)$ | **0** | **Low** | $O(sP)$ | $O(sP)$ |
| **SparsyFed** | $O(s \cdot P)$ | $O(EB \cdot P)$ | None | $O(sP) + \text{Idx}$ | $O(sP) + \text{Idx}$ |
| **DisPFL** | $O(s \cdot P)$ | $O(P \log P)$ | None | $O(sP) + \text{Idx}$ | $O(sP) + \text{Idx}$ |
| **FLASH-JMWST** | $O(s \cdot P)$ | $O(P \log P)$ | High | $O(sP) + \text{Idx}$ | $O(sP) + \text{Idx}$ |
| **FLASH-SPDST** | $O(s \cdot P)$ | **0** | **High** | $O(sP)$ | $O(sP)$ |

*FLASH-SPDST/JMWST require a "Warm-up" Stage ($E_d$ epochs training), whereas SSFL is single-shot or constant minibatch.

## A.4 Communication Encoding Overheads

In the main body, we adopt the standard *values-only encoding* assumption (counting only non-zero weight values, not their structural indices), consistent with prior work on sparsity and federated optimization (Dai et al., 2022; Liu et al., 2025; Lin et al., 2018). This isolates the savings due to sparsity itself, while recognizing that efficient communication and encoding schemes are an orthogonal line of work (Wen et al., 2017; Bernstein et al., 2018). Here, we quantify the specific overheads of dynamic topology transmission.

Let $P$ be the number of model parameters and $s \in (0, 1]$ be the **density** (fraction of non-zero elements). A dense model transmitted in `fp32` requires:

$$\text{Cost}_{\text{dense}} = 4P \quad \text{bytes.}$$

For sparse models, dynamic methods must transmit the topology. Under coordinate (COO) encoding, the cost for $k = sP$ non-zeros is:

$$\text{Cost}_{\text{COO}} = 4k \text{ (vals)} + 4k \text{ (idxs)} = 8sP \quad \text{bytes.}$$

Bitmask encoding sends a dense binary mask of length $P$ alongside the values:

$$\text{Cost}_{\text{bitmask}} = 4k + \frac{P}{8} = 4sP + 0.125P \quad \text{bytes.}$$

Dynamic sparsity methods (e.g., DisPFL, SparsyFed) must pay one of these overheads *at every round* to update the mask. In contrast, SSFL discovers a global mask once at initialization. After this one-time broadcast, the topology is static, and clients strictly transmit values:

$$\text{Cost}_{\text{SSFL}} = 4k = 4sP \quad \text{bytes.}$$

Table 5 compares these costs. At moderate density ($s = 0.5$), SSFL matches the theoretical limit of sparse compression. Crucially, at high sparsity ($s = 0.05$), SSFL achieves the theoretical minimum cost (5.0%), outperforming bitmasks (8.1%) which are limited by fixed overheads, and maintaining a 2× advantage over COO encoding (10.0%).

Table 5: Communication cost (bytes) under different encoding schemes. SSFL achieves the minimal theoretical cost ($4sP$) by eliminating the recurring structural overhead required by dynamic methods.

| Encoding scheme | Formula | Cost at $s = 0.5$ | Cost at $s = 0.05$ |
|---|---|---|---|
| Dense | $4P$ | 100.0% | 100.0% |
| Bitmask (Dynamic) | $4sP + P/8$ | 53.1% | 8.1% |
| COO (Dynamic) | $8sP$ | 100.0% | 10.0% |
| **SSFL (Fixed Mask)** | $4sP$ | 50.0% | 5.0% |

### A.5 Static Sparsity and Hardware Acceleration

A critical advantage of SSFL's static masking approach is its compatibility with hardware acceleration on modern AI accelerators. Static sparsity patterns enable substantial performance gains that are difficult or impossible to achieve with dynamic sparse training methods.

**Hardware Support for Static Sparsity.** Modern AI accelerators increasingly provide native support for sparse operations with fixed patterns. NVIDIA's Ampere and Hopper architectures introduced Sparse Tensor Cores (Mishra et al., 2021) that exploit 2:4 structured sparsity to achieve 2× math throughput compared to dense operations. Similarly, Graphcore's Intelligence Processing Units (IPUs) provide optimized kernels for static block sparsity (Li et al., 2023), demonstrating that static sparse implementations can outperform dense equivalents at 90% sparsity. These hardware optimizations rely fundamentally on knowing the sparsity pattern at compile time, enabling efficient memory layout, vectorized operations, and optimized instruction scheduling (NVIDIA, 2020b).

**Why Static Masks Enable Acceleration.** Static sparsity patterns provide three key advantages for hardware acceleration: (1) Compile-time optimization: fixed masks allow compilers to generate optimized code with predictable memory access patterns and efficient sparse data structures (Gale et al., 2020); (2) Zero runtime overhead: the mask need not be recomputed, compressed, or communicated during training, eliminating dynamic topology management costs (PyTorch Team, 2024); and (3) Hardware-friendly formats: static patterns can be stored in compressed formats (e.g., CSR, blocked layouts) that align with specialized hardware instructions (NVIDIA Corporation, 2023b;a). For instance, NVIDIA's cuSPARSELt library achieves near-linear speedup with sparsity when using static patterns, as the compressed representation can be pre-computed and the metadata efficiently organized for Tensor Core operations (NVIDIA Corporation, 2022).

**Limitations of Dynamic Sparsity for Hardware Acceleration.** In contrast, dynamic sparse training methods face fundamental barriers to hardware acceleration. Dynamic methods must repeatedly prune and regrow connections during training, which introduces several sources of overhead: (1) Runtime mask recomputation: dynamic methods incur $O(P \log P)$ sorting costs or $O(B \cdot P)$ per-batch costs for topology

updates Evci et al. (2020b); Guastella et al. (2025), where $P$ is the parameter count and $B$ is the number of batches; (2) Control-flow bottlenecks: changing sparse patterns break vectorization and prevent efficient use of specialized instructions Hubara et al. (2021); Lasby et al. (2023); (3) Memory access irregularity: dynamic topology changes lead to unpredictable, scattered memory accesses that underutilize memory bandwidth Gale et al. (2020); and (4) Metadata overhead: dynamic methods must repeatedly compress, decompress, and communicate mask indices, often requiring metadata storage that exceeds the weight storage itself Pool (2021).

Empirical evidence confirms these limitations. Hardware studies show that dynamic sparse patterns fail to achieve practical speedups on GPUs and specialized accelerators Lasby et al. (2025); Curci et al. (2024). Neuromorphic computing research demonstrates that static masks achieve up to $2.06\times$ better energy-delay product than dynamic masks specifically because they eliminate additional computation and data movement Soni et al. (2025). Even when dynamic methods achieve theoretical FLOP reductions, these do not translate to wall-clock speedups due to the overhead of mask management Graphcore (2023).

**Compatibility with Accelerators.** By discovering a global mask once and maintaining it throughout training, SSFL's design allows it to be compatible with the same hardware optimizations available for sparse inference, during the local training epochs. The fixed mask can be compressed offline, stored in hardware-optimized formats, and exploited by specialized kernels on GPUs (via cuSPARSE/cuSPARSELt), TPUs, and custom accelerators Nakahara et al. (2019); Thangarasa et al. (2023). This positions SSFL not only as communication-efficient but also as a pathway to training-time acceleration on next-generation sparse-optimized hardware, an advantage that dynamic methods cannot easily match.

### A.6 Adaptation to Out-of-Distribution (OOD) Shifts

Resilience to non-stationary environments, such as the arrival of clients with novel data classes, is critical for robust federated learning. While SSFL focuses on a stable global sparsity for efficiency, the framework allows for adaptation through *OOD Adaptation* when new clients with different data distribution joins the federated learning scheme. This approach triggers mask rediscovery when distribution shifts are detected, avoiding the instability and overhead of continuous dynamic updates.

---

**Algorithm 2** SSFL Extension: OOD Adaptation

---

**Require:** Clients $\mathcal{K}_{\text{total}}$, current model $\boldsymbol{w}_r$, current mask $\mathbf{m}$, sparsity level $\sigma$
**Ensure:** Updated mask $\mathbf{m}_{\text{new}}$ and adapted model $\boldsymbol{w}_r$
1: *// Note: $\boldsymbol{w}_r$ contains trained values for active params and init values for pruned params*
2:
3: *// Phase 1: Single-Shot Saliency Re-aggregation*
4: **for** each client $k \in \mathcal{K}_{\text{total}}$ in parallel **do**
5:     Sample minibatch $B_k \subset \mathcal{D}_k$
6:     Compute local saliency $\boldsymbol{s}_k$ using Equation (1) on $\boldsymbol{w}_r$
7:     Send $\boldsymbol{s}_k$ and $n_k = |\mathcal{D}_k|$ to server
8: **end for**
9:
10: *// Phase 2: Server-Side Mask Update*
11: Server computes data proportions: $p_k = n_k / \sum_{i=1}^K n_i$
12: Server aggregates global saliency: $\boldsymbol{s}_{\text{global}} \leftarrow \sum_{k=1}^K p_k \boldsymbol{s}_k$
13: Server generates new mask: $\mathbf{m}_{\text{new}} \leftarrow \text{TOPK}(\boldsymbol{s}_{\text{global}}, \lfloor (1-\sigma) \cdot d \rfloor)$
14:
15: *// Phase 3: Update Model State*
16: $\mathbf{m} \leftarrow \mathbf{m}_{\text{new}}$
17:                   ▷ New active weights automatically start at init values present in $\boldsymbol{w}_r$
18: Broadcast updated mask $\mathbf{m}$ to all clients
19: **return** $\mathbf{m}$, $\boldsymbol{w}_r$

---

### A.6.1 Algorithm Extension for OOD adaptation

We formalize our OOD adaptation in Algorithm 2. When OOD clients enter at round $r_{\text{intro}}$, the server triggers a single mask update. Clients compute saliency scores using the full local model state $\boldsymbol{w}_r$, effectively "unmasking" the pruned parameters which retain their original initialization values in memory. This allows the sensitivity analysis to evaluate both the currently active trained weights and the potential utility of inactive connections for the new data.

The aggregated saliency vector $\boldsymbol{s}_{\text{global}}$ identifies the optimal subnetwork for the combined distribution. The mask is updated to $\mathbf{m}_{\text{new}}$, activating the new high-saliency connections. Since $\boldsymbol{w}_r$ preserves the initialization values for these previously pruned parameters, training simply resumes with the new mask, ensuring a stable optimization landscape without complex restoration steps.

**Experiment setup.** For a proof of concept, we designed a controlled distribution shift experiment using the CIFAR-10 dataset, partitioned into two disjoint tasks: an In-Distribution (ID) task consisting of classes 0–7, and an Out-of-Distribution (OOD) task consisting of the held-out classes 8–9 (Ship and Truck). The training timeline was divided into two phases. In the *Initial Phase* (Rounds 0–225), only ID clients participated, ensuring the model had zero exposure to the OOD classes. At Round 225, we introduced the OOD clients and triggered a single execution of our mask discovery algorithm to adapt the global sparse topology to the new data distribution before resuming training.

**Alternative Adaptation Strategies and Future Work.** While in this preliminary exploration of OOD adaptation we focus on trigger-based adaptation for efficiency, other strategies within the SSFL framework are possible. A *periodic refinement* approach could re-evaluate saliency at fixed intervals rather than waiting for a specific trigger, offering continuous adaptation at the cost of higher cumulative communication overhead. Alternatively, an *incremental growth* strategy could strictly retain the existing mask and only activate *additional* parameters for OOD data (increasing total density over time). This is possible due to the simplicity and flexibility of our approach. We leave such explorations for future work.

### A.7 Analysis of Warm-up and Mask Discovery at Initialization

A natural question regarding pruning-at-initialization is whether the gradient-based saliency metric defined in Equation (1) is overly sensitive to the random initialization of parameters $w_0$. Intuitively, a "warmup" phase of dense training could stabilize weights and potentially yield more informative gradients. To investigate this in the Federated Learning setting, we conducted a rigorous empirical comparison between our standard SSFL (mask discovery at initialization, $t = 0$) and *Warmup-SSFL*, which performs 10 rounds of dense federated training before computing saliency and generating the mask.

**Empirical Results.** We evaluated both methods on CIFAR-10 with ResNet-18 across a spectrum of sparsity levels (50%–95%). To ensure a fair comparison, both methods were evaluated at the same total after training round (step 500). As shown in Table 6, the performance difference between initialization-based discovery and post-warmup discovery is negligible, with deltas ranging from $-0.01\%$ to $+0.99\%$. We leave rigorous analysis of the effect of warm-up for future work.

Table 6: Comparison of SSFL (mask discovery at initialization) vs. Warmup-SSFL (mask discovery after 10 dense rounds) on CIFAR-10. All numbers are percentages.

| Sparsity | SSFL (At Init) | Warmup-SSFL | Difference |
|:---:|:---:|:---:|:---:|
| 95 | 83.57 | 83.12 | +0.45 |
| 90 | 85.10 | 84.67 | +0.43 |
| 80 | 85.35 | 86.34 | -0.99 |
| 70 | 87.02 | 87.32 | -0.30 |
| 50 | 88.29 | 88.30 | -0.01 |

**Explanation in light of Theoretical Context and Lottery Tickets.** Our findings are consistent with the *Lottery Ticket Hypothesis* literature (Frankle & Carbin, 2019; Frankle et al., 2019; 2020a; Morcos et al., 2019; Chen et al., 2021), specifically the work on *Stabilizing the Lottery Ticket Hypothesis* (Frankle et al., 2020b) which provides a potential explanation. Frankle et al. (2019) demonstrate that for networks of moderate depth (e.g., ResNet-18) on datasets like CIFAR-10, winning lottery tickets can be successfully identified at initialization (iteration 0). It is primarily for significantly deeper networks or larger datasets (e.g., ResNet-50 on ImageNet) that "late rewinding" or resetting weights to an early training iteration rather than initialization becomes necessary to find stable subnetworks. Our experimental setting falls within the regime where initialization-based pruning is theoretically and empirically robust.

**The Effect of Non-IID Data.** In the specific context of Federated Learning, we hypothesize that warmup strategies face an additional challenge: *client drift* induced by data heterogeneity. During early dense training rounds on non-IID data, local models may drift significantly toward their local distributions. Saliency scores computed after this drift risk capturing parameters important for local overfitting rather than global generalization. Aggregating these biased scores may cancel out the benefits of weight stabilization. In contrast, the random initialization used in SSFL provides a neutral starting point, potentially allowing the aggregated gradient saliency to capture structural importance unbiased by local data quirks.

**Conclusion.** Given that (1) performance differences are negligible, (2) literature supports initialization-based pruning for this model scale, and (3) warmup incurs a high communication cost (transmitting dense models for initial rounds), we opted for single-shot discovery at initialization for its optimal efficiency-accuracy trade-off for SSFL. Future work scaling SSFL to very large foundation models may revisit this trade-off, potentially leveraging techniques like warm-up as model capacity grows.

## A.8  Experimental settings and further results

The performance of our proposed method is assessed on three image classification datasets: CIFAR-10, CIFAR-100 Krizhevsky et al. (2009), and Tiny-Imagenet. We examine two distinct scenarios to simulate non-identical data distributions among federating clients. Following the works of Hsu et al. (2019), we use Dir Partition, where the training data is divided according to a Dirichlet distribution $\text{Dir}(\alpha)$ for each client, and the corresponding test data for each client is generated following the same distribution. We take an $\alpha$ value of 0.3 for CIFAR-10, and 0.2 for both CIFAR-100 and Tiny-Imagenet. Additionally, we conduct an evaluation using a pathological partition setup, as described by Zhang et al. (2020), where each client is randomly allocated limited classes from the total number of classes. Specifically, each client holds 2 classes for CIFAR-10, 10 classes for CIFAR-100, and 20 classes for Tiny-Imagenet in our setup, similar to Dai et al. (2022).

**Experiment Details and hyper-parameters**  As outlined in Algorithm 1, the client models are trained on their local data and only during a communication round $r$, the randomly selected clients in $\mathcal{C}'$ share their weights for aggregation. During the local training, we use the SGD optimizer for SSFL and also for all baseline techniques, employing a weighted decay parameter of 0.0005. With the exception of the Ditto method, we maintain a constant of 5 local epochs for all methods. However, for Ditto, in order to ensure equitable comparison, each client undertakes 3 epochs for training the local model and 2 epochs for training the global model. Our initial learning rate stands at 0.1 and diminishes by a factor of 0.998 after each round of communication, similar to Dai et al. (2022). Throughout all experiments, we use a batch size of 16 due to the usage of group normalization Wu & He (2018). We execute $R = 500$ global communication rounds for CIFAR-10, CIFAR-100 and Tiny-ImageNet. For the experiments in Table 1 and Table 2, we use a sparsity level $s = 50\%$ similar to Dai et al. (2022) for fair comparison. We however, include experiments with increasing sparsity levels up to $s = 95\%$ and compare SSFL with other sparse FL methods, including a baseline random-masking on non-IID data, in Figure 2.

For FedPM, we use their official implementation Isik et al. (2022) and extend it to support ResNet architectures and the CIFAR-100 dataset to ensure fair comparisons. For SparsyFed, we likewise rely on the official implementation Guastella et al. (2025) and re-run their codebase under our general experimental setup, matching our communication rounds ($R = 500$) and Dirichlet $\alpha$ parameters. These $\alpha$ values follow the settings commonly adopted in prior work, including Dai et al. (2022), to provide consistency across baselines.

For all baselines, we used the authors' official implementations and recommended hyperparameters, without additional tuning. SSFL likewise required no hyperparameter tuning beyond setting the target sparsity level, ensuring a fair comparison across methods.

**Model Initialization** All models in our experiments were randomly initialized using the standard Kaiming (He) Normal initialization scheme (He et al., 2015) for convolutional layers, which helps maintain variance stability in deep networks. The final fully connected layer used PyTorch's default Kaiming Uniform initialization. Batch normalization layers were initialized with weights of 1 and biases of 0. This random initialization defines the starting state $w_0$ from which local saliency scores are computed in the first round, ensuring no prior task-specific information is encoded in the model weights before the saliency aggregation step. Across 5 independent runs with different random seeds, SSFL exhibited a standard deviation of $< 0.4\%$ in final accuracy, demonstrating that the saliency signal extracted at initialization is consistent and robust to the specific random draw of weights.

**Implementation of Baselines:** We describe most of the implementation detail for the baselines in Section 4. We use the official implementaion of DisPFL by Dai et al. (2022) available on their github page and use similar hyper-parameters for both DisPFL and SSFL. FedPM trains sparse random masks instead of training model weights and finds an optimal sparsity by itself Isik et al. (2022). We use the official implementation of FedPM available in their github page for the paper. For fair comparison, we use the CIFAR-10 non-IID split that we use in this work with the $\alpha$ values as described in Appendix A.2. For random-$k$ baselines, we follow a training strategy that is similar to SSFL and in essense to FedAvg McMahan & Ramage (2017) or DisPFL Dai et al. (2022) in terms of the weight aggregation mechanism. To establish random-$k$ as a random sparse training, similar to our proposed sparse FL and DisPFL training method we generate a random-$k$ mask at each local site and train and aggregate the weights corresponding to that mask after every $T$ steps of training.

**top-$k$** Top-k is a popular choice for distributed training with gradient sparsification Barnes et al. (2020); Lin et al. (2017) and also in variation of federated learning where gradient aggregation instead of weight aggregation takes place. We, however, did not find works that employ top-$k$ sparsification in the weight-averaging scheme of Federated Averaging. As a result, to test this we implemented a top-k baseline, where the top-k weights from each local model at the end of $T$ steps are shared among the selected clients in each communication round and are aggregated. We note that unlike SSFL, DisPFL or random masking, at the end of training top-$k$ does not result in sparse local models with the benefits of sparse models such as faster inference.

**Convergence under longer training and more communication rounds** We report the convergence plots for SSFL and other methods in Figure 7 (a-b) when trained for a total of $R = 500$ communication rounds. As Ditto does not properly converge within 500 rounds, we conduct further experiments for SSFL, DisPFL and Ditto on a much longer $R = 800$ communication rounds to analyze the convergence of these methods. We demonstrate the convergence plots in Figure-7 (c).

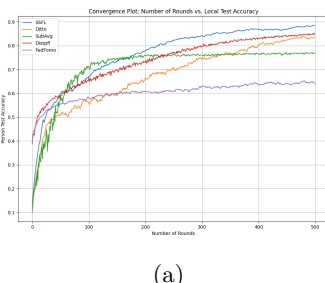 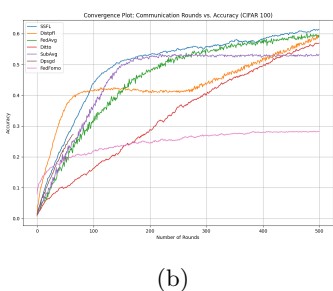 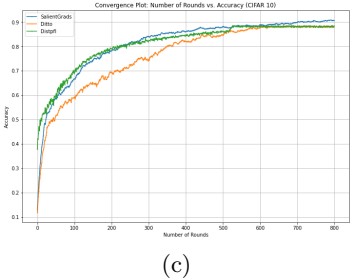

(a)                                      (b)                                      (c)

Figure 7: Comparison of convergence rates of SSFL with similar federated and sparse federated learning methods. (a) CIFAR-10 with $R = 500$ rounds, (b) CIFAR-100 with $R = 500$ rounds, (c) longer training $R = 800$ rounds.

### A.9 Quality of the discovered mask across heterogeneous clients

Figure 4 (c-d) provides an analysis of how the global mask discovered by SSFL transfers across clients with highly imbalanced data availability. Figure 4 (c) shows the test accuracy of local sparse models trained with the global mask, along with the weighted score assigned to each client. Figure 4 (d) shows the corresponding distribution of training and test samples across clients.

Despite substantial differences in the amount of data available at each site, the global mask enables consistently strong performance across the federation. Clients with very limited data still achieve reasonable accuracy, while high-data clients maintain competitive performance. This demonstrates that the aggregated saliency scores produce a global mask that generalizes across diverse clients, rather than overfitting to any single site. Importantly, this analysis highlights that even clients contributing relatively little data can benefit from the shared sparse subnetwork.

### A.10 Local Client Accuracy and Class Distribution

We plot the class distribution with $\text{Dir}(\alpha)$ on the CIFAR10 dataset with $\alpha = 0.3$ for 10 random clients and their final accuracy in Fig 8 (a). In Fig (b) we plot the top-10 clients in terms of their final test accuracy and in Fig-(c) the bottom 10 clients in terms of final test accuracy. We notice that, SSFL finds masks that result in consistent local model performance and even in the bottom 10 clients, the performance remains respectable.

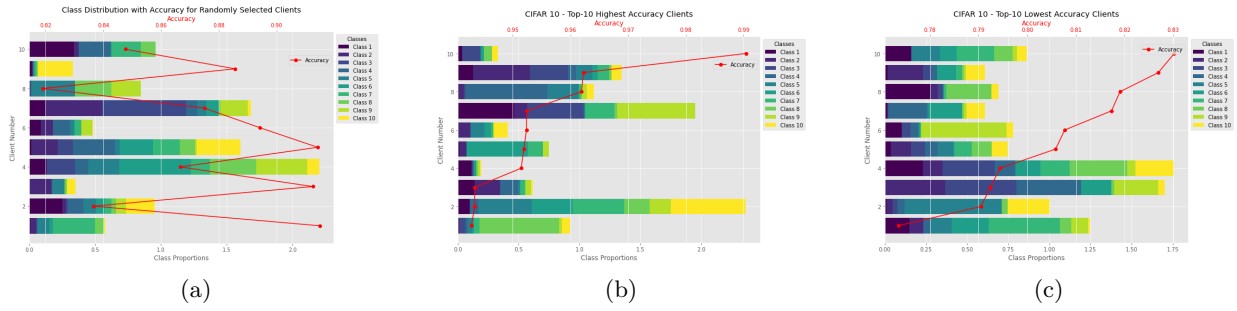

(a)          (b)          (c)

Figure 8: (a) Class distribution with Dir(0.3) for the CIFAR-10 dataset for 10 random clients and their final local test accuracy (b) the top-10 clients in terms of their final test accuracy (c) the bottom-10 clients in terms of their final test accuracy

## B  Extended Literature Review

For completeness, we provide a broader overview of related work, complementing the focused discussion in Section 2 of the main text. This section is not required to follow our method or experiments but may provide useful background for readers seeking a more complete survey of related areas. We group this discussion into:

(i) neural network pruning, (ii) sparsity and efficiency in federated learning, (iii) pruning at initialization, and (iv) pruning at initialization in federated learning.

**Neural Network Pruning**  Like most areas in deep learning, model pruning has a rich history and is mostly considered to have been explored first in the 90's (Janowsky, 1989; LeCun et al., 1990; Reed, 1993). The central aim of *model pruning* is to find subnetworks within larger architectures by removing connections. Model pruning is very attractive for a number of reasons, especially for real-time applications on resource-constraint edge devices which is often the case in FL and collaborative learning. Pruning large networks can significantly reduce the demands of inference Elsen et al. (2020) or hardware designed to exploit sparsity Cerebras (2019); Pool et al. (2021). More recently the *lottery ticket hypothesis* was proposed which predicts the existence of subnetworks of initializations within dense networks, which when trained in isolation from scratch can match in accuracy of a fully trained dense network Frankle & Carbin (2019).

This rejuvenated the field of sparse deep learning Renda et al. (2020); Chen et al. (2020) and more recently the interest spilled over into sparse reinforcement learning (RL) as well Arnob et al. (2024; 2021); Sokar et al. (2021). Pruning in deep learning can broadly be classified into three categories: techniques that induce sparsity before training and at initialization Lee et al. (2018); Wang et al. (2020); Tanaka et al. (2020), during training, including methods based on magnitude pruning Zhu & Gupta (2018), weight transformations Ma et al. (2019), regularization-based approaches Yang et al. (2019), explicit sparse projections Ohib et al. (2022); Ohib (2023); Ohib et al. of the matrices and post-training Han et al. (2015); Frankle et al. (2021).

The advent of large language models (LLMs) has further amplified interest in pruning and sparsity techniques, as these models often contain billions of parameters that pose significant computational and memory challenges for deployment. Recent work has explored various pruning strategies for LLMs, including magnitude-based pruning Frantar & Alistarh (2023), structured pruning to remove entire attention heads or layers Ma et al. (2023), and semi-structured sparsity patterns that balance compression with hardware efficiency Sun et al. (2023). Additionally, techniques such as quantization-aware pruning and knowledge distillation have been combined with sparsity to achieve extreme compression ratios while maintaining model performance Xiao et al. (2023); Kurtić et al. (2023). Sparsity has also been leveraged in parameter-efficient fine-tuning through sparse adapters Arnob et al. (2025), low-rank adaptations Hu et al. (2022); Zhang et al. (2023), and mixture-of-experts architectures that activate only subsets of parameters per input Fedus et al. (2022); Jiang et al. (2024). These advances demonstrate that pruning remains crucial for making state-of-the-art models practical for resource-constrained environments, extending from traditional computer vision tasks to modern natural language processing applications.

**Sparsity, pruning and efficiency in FL**   Federated Learning (FL) has evolved significantly since its inception, introducing a myriad of techniques to enhance efficiency, communication, and computation. Interest in efficiency has increased further with the proposal of the Lottery Ticket Hypothesis (LTH) by Frankle & Carbin (2019). However, identifying these tickets has traditionally been challenging. In the original work, the strategy employed to extract these subnetworks was to iteratively train and prune the network until the target sparsity was attained. However, this process is unsustainable in the FL setting and would in contrast make the training even more expensive due to the significant increase in training duration. Variations of this idea have been employed in the FL setting, as demonstrated by Li et al. (2020); Jiang et al. (2022); Seo et al. (2021); Liu et al. (2021) and Babakniya et al. (2022). However, many of them either suffer from the same issue of iterative pruning and retraining which is extremely costly or decline in performance when this train, prune and retrain cycle is skipped.

Pruning and sparsity are highly advantageous in federated learning (FL), especially since many applications involve clients with limited resources. As sparse models can be substantially efficient during inference (Li et al., 2016), training sparse local models in resource constrained local sites could be an effective approach. In FL setting, pruning has been explored with mixed success, often employing a range of heuristics. Works, such as by Munir et al. (2021) focused on pruning resource-constrained clients, Yu et al. (2021) on using gated dynamic sparsity, Liu et al. (2021) performs a pre-training on the clients to get mask information, and Jiang et al. (2022) relies on an initial mask selected at a particular client, followed by a FedAvg-like algorithm that performs mask readjustment every $\Delta R$ rounds. However, they either do not leverage sparsity fully for communication efficiency, suffer from performance degradation or is not designed for the challenging, realistic non-IID setting.

Works by Dai et al. (2022) and Bibikar et al. (2022) utilize dynamic sparsity similar to Evci et al. (2020a) and extends it to the FL regime, however, they both start with random masks as an initialization. A recent focus has also been placed on random masking and optimizing the mask Setayesh et al. (2022) or training the mask itself instead of training the weights at all Isik et al. (2022). Furthermore, in light of the recent surge in the development of large language models (LLMs), the exploration of federated learning in this domain is taking place, although relatively recently Fan et al. (2023); Yu et al. (2023). Investigating sparsity and efficiency within this context presents a fascinating and promising avenue for research.

**Pruning at Initialization**   The work on the *lottery ticket hypothesis* (LTH), which showed that from early in training and often at initialization, there exist subnetworks that can be trained in isolation to full accuracy,

opened up the prospect for pruning at initialization (PaI). Several methods have been proposed for pruning at initialization such as SNIP (Lee et al., 2018), which aims to prune weights that are least important for the loss, GraSP (Wang et al., 2020), which prunes weights aiming to preserve gradient flow and SynFlow (Tanaka et al., 2020) aims to iteratively prune weights with the lowest "synaptic strengths". These methods have been later studied in depth by Frankle et al. (2021), where they provide interesting insights about the efficacy of these methods, including the observation that SNIP Lee et al. (2018), which is a variation of the gradient based connection saliency criterion (Mozer & Smolensky, 1988), consistently performs well. Hence, we choose gradient based connection saliency as the basis for our method.

Frankle et al. (2021) found that PaI methods tend to prune networks at the layer level rather than the connection level in single-node deep learning scenarios. In Section 4, we explore the applicability of this observation to non-IID federated learning settings

**Pruning at initialization in FL**   We did not discover any study that directly employs connection importance at initialization in the Federated Learning (FL) setting. Nevertheless, research such as Jiang et al. (2022) incorporated gradient-based importance criterion, SNIP Lee et al. (2018); De Jorge et al. (2020), within their comparative benchmarks, but calculating such scores only on the local data of one particular client site in the FL setting, resulting in subpar performance. Similarly, Huang et al. (2022) presents connection saliency in their benchmark but assumes that to calculate meaningful parameter importance scores, the server has access to a public dataset on which such saliency scores can be calculated. However, that is in general unlikely to happen and would lead to privacy concerns if such constraints were enforced.

In this work, we demonstrate that it is possible to find sparse masks or sub-networks through gradient based connection saliency measures at initialization, considering the distribution of training data at local client sites. Since, to the best of our knowledge we did not find any comparable pruning at initialization methods in the non-IID FL setting, we compare our method to approaches such as Dai et al. (2022) that results in sparse local models and similar sparse and dense baselines in Section 4. A variation of our work has also been applied in the domain of federated distributed neuroimaging (Thapaliya et al., 2024), where gradient averaging, first demonstrated in (Ohib et al., 2023) in this setup, was used for the FL process instead of model averaging.

