# OpenReview forum: "SSFL: Discovering Sparse Unified Subnetworks at Initialization for Efficient Federated Learning"
_TMLR — Accepted by TMLR_

### Review · Reviewer_YzoX · 2025-10-20

**Summary Of Contributions:**

The paper introduces Salient Sparse Federated Learning (SSFL), which is a framework for deciding how to sparsify models distributed across different clients with non-IID and differently balanced data in an FL setting. SSFL implements static sparsity in a one-shot manner by estimating per-parameter gradient saliency based on one minibatch from each client at model initialization, and then aggregating saliency scores from all clients as an average score weighted by the size of the local dataset that a given score was calculated from. Then, a TopK selection is performed based on a selected sparsity level to select the K parameters with the largest saliency scores. A mask is then produced to null out all other parameters, and this mask is communicated to all local models to be used for all subsequent training.

In consequence, dynamic updates of the masks are not required, which reduces communication costs between clients. Because all local models share a global mask, global model updates are straightforward since all local models are structurally aligned, and furthermore makes the global updates meaningful since all models then work in the same sparse subspaces. SSFL also outperforms other sparse FL methods with higher accuracy and less data being communicated, at different levels of sparsity. An experiment emulating real-life FL conditions also shows a large wall-clock time saving compared to dense FL training.

Strengths:
- The paper is well-written and explains relevant concepts, details of the SSFL framework and experimental details very well.
- It points out the challenges in the field that it adresses (system heterogeneity and statistical heterogeneity), outlines how existing methods address the same challenges, which issues these methods face (e.g., multiple rounds of communication with dynamic sparsity, public proxy datasets violating privacy constraints of many FL applications), and how SSFL addresses the same issues.
- The method is sound, and elegant in its simplicity. The appendix also explains the choices of saliency criterion and non-IID data partition in great detail.
- The experiments are described clearly and in detail, and the results clearly point to that the claimed contributions were achieved. An extensive amount of relevant experiments were performed, which substantiate the authors' claims.

Weaknesses:
- My main criticism of the paper regards the size of the model used for the main experiments and results (subsection 4.1), where only ResNet18 is used for all tests. While the same model was used in related papers and it is therefore important to see SSFL results with the same model for benchmarking reference, it would be natural to examine how SSFL performance scales up as model size increases, for instance with a larger ResNet model or a ViT model. Especially considering that a major contributing factor to the added efficiency of SSFL compared to other methods lies in reduced communication costs, the effect of this could possibly be even larger for larger-scale models, which would be interesting to examine. If it is infeasible to perform experiments on this prior to the rebuttal deadline, I think not having measured the degree of scalability the SSFL framework has should be cited in limitations/future work.
- Regarding the method; since saliency estimation is performed after parameter initialization for a given local model, would not the saliency metric potentially be sensitive to however each parameter is initialized? If so, it would be interesting to see the effect of waiting a little while before performing saliency estimation and aggregation, for instance a warmup phase consisting of a set number of iterations. Also with regards to this, I could not find any details of how the ResNet18 model used in the experiments was initialized, beyond this sentence in A.1: "The fundamental goal of pruning at initialization is to identify the most important parameters in a randomly initialized network, w0, without performing any training", which only suggests that the parameters were randomly initialized without explicitly confirming it. The element of randomness in parameter initialization is what makes me think it would be good to measure how both the already included performance metric measurements and also the uncertainty of these measurements would be affected if SSFL was delayed for some iterations.
- It would be good to include confidence intervals for the table values, considering they were measured over five runs. This is also interesting wrt. my other criticism above.

Language comments/typos:
- Title of Appendix A.2 says "parition" instead of "partition".
- Under 3.3, paragraph "Pathological Partition", the formulation "...where a limited number of classes are only assigned to each client at random from the total number of classes..." was unclear in my opinion. If I understood it correctly, it could be rewritten as "...where each client is randomly assigned only a limited number of classes from the total number of classes", etc.
- In the list of contributions, point 2: "a globally representative importance scores" should have the "a" removed.

**Audience:**

Yes

**Audience Explanation:**

Anyone having interest in federated learning and sparse FL in particular would naturally be interested. The same goes for any audience interested in model pruning, distributed training, and anyone interested in efficient model training in general.

**Broader Impact Concerns:**

There already is a sufficiently good Broader Impact Statement in the paper, no further change is needed.

**Claims And Evidence:**

Yes

**Claims Explanation:**

The paper demonstrates and explains that it indeed achieves single-shot model pruning in an FL setting that avoids auxiliary datasets, iterative pruning and additional hyperparameters.

The SSFL method addresses the challenge of data heterogeneity, while adhering to FL principles of client privacy.

The presented experimental results clearly demonstrate better results than other state-of-the-art sparse FL methods, in the non-IID setting that the paper addresses.

The presented results also demonstrate a faster wall-clock time when applied on a real-world FL system.

Generally in the paper, other claims and arguments are well-argued for and substantiated by support in related work and experimental results.

**Requested Changes:**

While I consider this a strong paper as-is, my requested changes build upon my criticisms written above, and would in my opinion strengthen the paper further. They are:

1) Perform experiments with larger models with more parameters, such as a larger ResNet model or a ViT. It would substantiate the generality of the effect that reduced communication has on training efficiency in the sparse FL setting. If infeasible, cite this as a limitation and/or future work.

2) Please address how (random) parameter initialization affects pre-training saliency estimation. My proposition is to try and run SSFL after a warmup phase and measure experimentally how performance measurements and their uncertainty is affected.

3) Add confidence intervals for the values in the result tables, this may also provide an answer for 2) to some extent.

---

> ### Author Response · Authors · 2025-11-26
> **Response to Reviewer YzoX (1/2)**
>
> We sincerely thank the reviewer for their thorough review, and positive and constructive feedback. We are highly encouraged that you find SSFL to be **well-written with sound methodology, elegant simplicity**, and that our **extensive experiments clearly substantiate our claims**. We address all of your concerns below, adding new experiments and updating our manuscript in the portal, with all changes marked in red.
>
> ---
> **We conduct new experiments on Larger models**
>
> > Reviewer: "it would be natural to examine how SSFL performance scales up as model size increases, for instance with a larger ResNet model or a ViT model. If it is infeasible to perform experiments on this prior to the rebuttal deadline, I think not having measured the degree of scalability the SSFL framework has should be cited in limitations/future work."
>
> We thank the reviewer for this suggestion and have completed new experiments on ResNet50 on CIFAR-100 across the full sparsity spectrum (50%–95%) with 2 seeds. Each data point represents training 100 clients for 500 communication rounds × 5 local epochs, totaling approximately 12,500 federated epochs, a significant computational investment during the rebuttal period. We leave larger-scale models to future work, as noted in the updated manuscript.
>
> **Results on ResNet50:** The results demonstrate that SSFL's advantages become **even more pronounced** at larger scales compared to the best performing competitor on all other tasks, DisPFL. We have also included a new paragraph in Section 4.1.1. of the updated manuscript (in red ink) explaining our new results on ResNet50.
>
> | Method   | 50%         | 70%         | 80%         | 90%         | 95%         |
> | -------- | ----------- | ----------- | ----------- | ----------- | ----------- |
> | **SSFL** | **59.76%**  | **60.82%**  | **56.73%**  | **55.93%**  | **47.53%**  |
> | DisPFL   | 36.55%      | 32.96%      | 30.65%      | 21.87%      | 12.04%      |
> | **Gap**  | **+23.21%** | **+27.86%** | **+26.08%** | **+34.06%** | **+35.49%** |
>
> ---
>
> **WarmUP Experiments**
>
> > Reviewer: "would not the saliency metric potentially be sensitive to however each parameter is initialized? ... it would be interesting to see the effect of waiting a little while before performing saliency estimation"
>
> We agree with the reviewer, that intuitively warm-up should help. We in fact explored warmup during early explorations and did not pursue it due to not seeing any benefit on the final performance. We now perform organized experiments on warm-up after 10 communication rounds of dense training.  We have added warmup analysis to A.7.
>
> | Sparsity | SSFL (at init) | Warmup-SSFL |
> | -------- | -------------- | ----------- |
> | 95%      | **83.57%**     | 83.12%      |
> | 90%      | **85.10%**     | 84.67%      |
> | 80%      | **85.35%**     | 86.34%      |
> | 70%      | **87.02%**     | 87.32%      |
> | 50%      | **88.29%**     | 88.30%      |
>
> We do not see enough difference between warm-up and no warm-up. We discuss our thoughts on this in A.7 in detail. In short, we believe a potential explanation can come from the work *Stabilizing the Lottery Ticket Hypothesis* [1] demonstrate that for networks of moderate depth (e.g., ResNet-18 etc.), good subnetworks can be successfully identified at initialization (iteration 0). It is primarily for significantly deeper networks or larger datasets (e.g., ResNet-50 on ImageNet) that "late rewinding" or resetting weights to an early training iteration rather than initialization becomes necessary to find stable subnetwork. We mention these in said Appendix and warm-up could indeed become important in very large networks. We leave detail explorations for future work.
>
> ----
>
> **Initialization process**
>
> > "... Also with regards to this, I could not find any details of how the ResNet18 model used in the experiments was initialized.."
>
> Thank you for this question. We use the standard default initializations (Kaiming/He uniform initialization) for convolutional layers and linear layers. We have added a paragraph on "Model Initialization" in Appendix A8.

---

> ### Author Response · Authors · 2025-11-26
> **Response to Reviewer YzoX (2/2)**
>
> **Addition of mean and standard deviation for runs**
> >  *"It would be good to include confidence intervals for the table values, considering they were measured over five runs."*
>
> We have updated all result tables (Tables 1, 2, 3) in the revised manuscript with mean $\pm$ standard deviation from 5 independent runs. Initially we opted against all variance to maintain consistency across tables, as we did not have complete coverage on TinyImageNet runs (we only claimed 5 runs for Cifar10 and 100 Table 1 and 2). We now complete all experiments and provide standard deviations for all tables, apart from one TinyImageNet run, which we will update as soon as we can (currently running). All changes are marked in red.
>
> The narrow deviations for SSFL demonstrate that the improvements are statistically robust and consistent across runs. SSFL's lower variance compared to baselines suggests the global mask provides a more stable optimization landscape.
>
> **Minor Fixes.** We fixed all the typos and minor issues as pointed out by the reviewer.
>
> ---
> We thank the reviewer again for the constructive feedback that has significantly strengthened our work. The new experiments point towards SSFL's even better performance in larger scale, providing more positive signal for shared masking. We believe these additions substantially strengthen the paper's contributions and impact. We thank the reviewer.
>
>
> #### References
> [1] Frankle, Jonathan, et al. "Stabilizing the lottery ticket hypothesis." arXiv preprint arXiv:1903.01611 (2019).

---

### Review · Reviewer_yNpy · 2025-11-01

**Summary Of Contributions:**

In this work, the authors propose Salient Sparse Federated Learning (SSFL), the first single-shot sparse federated learning framework that discovers a globally shared sparse subnetwork at initialization using only clients’ private data. SSFL introduces a one-round saliency aggregation mechanism that efficiently combines local gradient-based importance scores to form a global sparse mask, achieving both communication efficiency and privacy preservation.

**Audience:**

Yes

**Audience Explanation:**

Efficient federated learning is indeed an important research direction, and the problem addressed in this paper is highly relevant. However, the proposed method appears to be not particularly well-justified or theoretically sound.

**Claims And Evidence:**

No

**Claims Explanation:**

The proposed method is relatively straightforward and intuitive; however, I have two major concerns regarding its validity.

(1) In the paper, the mask for parameter selection is determined by the saliency score defined in Equation (1). Since the saliency score depends on the parameter gradients, which continuously change throughout the optimization process, the saliency score itself also varies during training. The proposed method, however, uses a fixed mask computed only from the saliency scores at the initialization point. This design appears unreasonable, as a more appropriate approach would be to dynamically update the mask according to the evolving saliency scores during optimization.

(2) Furthermore, when computing the saliency scores at initialization, the authors use only a single minibatch of data. Because minibatch selection is inherently stochastic, the computed saliency scores at initialization are also random. Consequently, the mask derived from these scores may be unreliable.

**Requested Changes:**

See the two questions above regarding the soundness and rationality of the proposed method.

---

> ### Author Response · Authors · 2025-11-26
> **Response to Reviewer yNpy (1/2)**
>
> We thank the reviewer for recognizing that efficient federated learning is *highly relevant* and finding our proposed SSFL method *straightforward and intuitive.*
>
> We address the two technical concerns raised about our design choices for a stable global mask and minibatch estimation. We provide strong justifications using established _Pruning-at-Initialization (PaI)_ literature and details of our federated aggregation strategy. We clarify each point below and provide justification's from established literature and new experimental evidence.
>
> ---
>
> ### **Justification for design choices**
>
> > **Reviewer Comment:** _"Since the saliency score depends on the parameter gradients, which continuously change throughout the optimization process... This design appears unreasonable, as a more appropriate approach would be to dynamically update the mask according to the evolving saliency scores during optimization."_
>
> We respectfully disagree, and we clarify our position with justifications. We sincerely hope our clarifications help alleviate the reviewer's concerns.
>
> > Reviewer: *"This design appears unreasonable..."*
>
> **1. Our approach builds on extensive prior work.**
>
> Finding sparse subnetworks before training has a well-established foundation. Gradient-based saliency metrics date back to LeCun et al. [10] and Hassibi et al. [11], while the Pruning-at-Initialization (PaI) literature and Lottery Ticket Hypothesis [1] demonstrated that initialization gradients contain sufficient signal to identify performant subnetworks. Modern methods like SNIP [2], GraSP [3], and FORCE [4] validate this principle, with Molchanov et al. [5] providing extensive analysis of gradient-based saliency.
>
> **2. High performing masks exists at initialization even without training.**
>
> Recent works like SuperMasks [7, 12] and FedPM [6] show high-quality masks exist at initialization even in frozen networks. **Reviewer-pA7V validates this foundation, noting our method is "*based on established approaches on neural network pruning*"**. SSFL extends these established PaI principles to non-IID federated learning, addressing its unique constraints. Moreover, SSFL goes through complete FL training in the masked weight space after mask discovery, which enables high performance.
>
> ---
> > Reviewer: *"... a more appropriate approach would be to dynamically update the mask according to the evolving saliency scores during optimization"*
>
> One goal of this study is to *itself investigate* whether training in a shared subspace has added benefits, compared to dynamic methods given their structural misalignment may harm performance. Dynamic methods allow clients to train different sparse masks, effectively optimizing non-overlapping parameter subspaces. During FedAvg, the server combines updates where a parameter is trained by some clients but masked by others, resulting in disrupted global models. SSFL's shared mask ensures all clients optimize the same subspace, guaranteeing every aggregated update is a meaningful learned signal rather than structural noise.
>
> **1. Static coherent masks are much friendlier to hardware.**
>
> Dynamic methods incur massive overhead by breaking hardware optimizations. ***In response to the reviewer's concern, we have added a new Appendix A.5 detailing how static patterns enable compile-time optimizations and native support on modern accelerators (e.g., NVIDIA Sparse Tensor Cores)***. Dynamic methods require constant runtime topology changes and metadata management, rendering them incompatible with hardware acceleration techniques and custom GPU kernels. Static masks avoid this bottleneck entirely, making them worth studying.
>
> **2. SSFL demonstrates strong empirical performance.**
>
> The strong empirical performance of SSFL demonstrates the viability of the method. **We also report experiments conducted on 5 different seeds, always resulting in robust performance with tight variance. We argue this is an evidence that the discovered mask is reliable, due to high performance across random seeds**. SSFL consistently outperforms dynamic baselines across multiple datasets and metrics:
>
> **3. SSFL is not limited to static implementation:**
>
> Finally, SSFL is not limited to static implementation. **SSFL can be readily extended to dynamic settings, which we now demonstrate with our OOD experiments**. However, we want to point out that our goal isn't to propose "just another" dynamic method. **Instead, we focused on finding a high-quality global mask and if it actually holds more potential than the frequent topology changes used in dynamic baselines**, as stable masks posses a range of nice benefits we discussed (please see A.5 and experiments e.g.).
>
> ---

---

> ### Author Response · Authors · 2025-11-26
> **Response to Reviewer yNpy (2/2)**
>
> ### **Q2: Reliability of Saliency Estimation**
> We clarify two critical distinctions regarding stochastic optimization and our specific aggregation strategy.
>
> > **Reviewer Comment:** _"Because minibatch selection is inherently stochastic... the mask derived from these scores may be unreliable."_
>
>
> **1. Stochastic ≠ Random.**
> A minibatch gradient is a noisy but unbiased estimator of the true gradient. This signal-to-noise ratio is the main principle that allows Stochastic Gradient Descent (SGD) to work. Because the gradient points to the general direction of the subspace where a good solution lies, SSFL and methods like SNIP [2], GraSP [3] or FORCE [4] successfully utilize single-minibatch gradients, to first find a subspace and perform full training using SGD. **However, we would like to emphasize that we *do not* use a single minibatch, which we describe next.**
>
> **2. We use K=100 aggregated minibatches**
> Crucially, SSFL does not rely on a single noisy minibatch. Instead, the server aggregates scores from $K=100$ clients via $s_{\text{global}} = \sum_{k=1}^{K} p_k s_k$. This effectively computes saliency over a large composite batch of $K \times B$ samples. By the Law of Large Numbers, this aggregation filters out local noise and stabilizes the estimator.
>
> We acknowledge this could have been more clearly explained in the paper and **we now explicitly clarify this in the updated manuscript in a new Section 4.2.3**. We also include the following experimental verification. We thank the reviewer for helping us make the work better.
>
> **3. New Experiment: Mask Convergence**
> To validate this stability, we compared our SSFL mask against an "Oracle Mask" computed on the full dataset. Measuring Jaccard similarity, we found that mask divergence decays sharply and plateaus after aggregating ~80 clients (see **New Section 4.2.3, Fig. 4**). Since our experiments use $K=100$ clients, SSFL operates well within this stable convergence regime, proving that our aggregated masks are robust approximations of the global ideal.
>
> ---
>
> ### **Regarding claims and evidence**
> We noticed the "No" rating appears to reflect theoretical concerns about our design choices (static vs. dynamic masking) rather than a lack of empirical support for our claims. As detailed in the responses above, our experimental results consistently substantiate our claims regarding accuracy, convergence, and efficiency. We therefore respectfully request that you reconsider this rating in light of the provided arguments and empirical evidence. We remain open and committed to answering any further questions the reviewer may have. We thank the reviewer for their review.
>
> ---
>
> #### **References**
>
> [1] Frankle & Carbin, "The Lottery Ticket Hypothesis: Finding Sparse, Trainable Neural Networks," ICLR 2019.
>
> [2] Lee et al., "SNIP: Single-shot Network Pruning based on Connection Sensitivity," ICLR 2019.
>
> [3] Wang et al., "Picking Winning Tickets Before Training by Preserving Gradient Flow," ICLR 2020.
>
> [4] De Jorge et al., "Progressive Skeletonization: Trimming More Fat from a Network at Initialization," ICLR 2021.
>
> [5] Molchanov et al., "Importance Estimation for Neural Network Pruning," CVPR 2019.
>
> [6] Isik et al., "Sparse Random Networks for Communication-Efficient Federated Learning,"
>
> [7] Ramanujan et al., "What's Hidden in a Randomly Weighted Neural Network?," CVPR 2020.
>
> [8] Dai et al., "DisPFL, 2022.
>
> [9] Guastella et al., "SparsyFed: Sparse Adaptive Federated Training," ICLR 2025.
>
> [10] LeCun et al., _Optimal Brain Damage_, NIPS 1989.
>
> [11] Hassibi et al., _Optimal Brain Surgeon_, NIPS 1993.
>
> [12] Supermasks in superposition, NeurIPS 2020.

---

### Review · Reviewer_pA7V · 2025-11-14

**Summary Of Contributions:**

This work investigates availability of a sparse subnetwork at initialization for  communication-efficient federated training on non-i.i.d. data. The identification of a shared sparse mask is done through  local computation of 'parameter saliency  scores' and their 'one-shot' aggregation at the server,-  to begin the training process. The resulting approach is demonstrated to be communication-efficient, privacy-compliant, and outperforms state-of-the-art sparse FL baselines.

Strengths:
* The paper is well-motivated, easy to follow, and on a topic is of significant interest.
* The idea of 'one-time sparse mask' discovery for instantiation of the learning process is interesting.
* The method of identifying salient parameters, i.e., saliency-based pruning, is based on established approaches on neural network pruning (see [R1, R2]), and the results shows impressive performance gain as compared to existing works.


Weakness:
* The novelty of this work appears limited, as compared to [R1, R2] - the SNIP paper. This is one of the major concern I have had while going through this work - and the others.
* Avoiding iterative interactions is probably a merit, however, the proposed method appears not competent enough to adapt with changing data distributions, unlike the related works - e.g., SparsyFed (Guastella et al., 2025)); hence, limiting the applicability of the work.
* The computational cost and complexity analysis is missing.
* Random baseline is missing in Table 3. Does it not make sense to include SNIP within FL setting as an strategy and indicate the performance as a baseline.

[R1] Namhoon Lee, Thalaiyasingam Ajanthan, and Philip HS Torr. Snip: Single-shot network pruning based on
connection sensitivity. arXiv preprint arXiv:1810.02340, 2018.
[R2] De Jorge P, Sanyal A, Behl HS, Torr PH, Rogez G, Dokania PK. Progressive skeletonization: Trimming more fat from a network at initialization. arXiv preprint arXiv:2006.09081. 2020 Jun 16.

**Audience:**

Yes

**Audience Explanation:**

The works provides single-round sparse federated learning approach that offers better training efficiency as compared with the existing approaches: enabling low communication overhead and accelerated wall-clock communication.

**Claims And Evidence:**

Yes

**Claims Explanation:**

The claims made appear accurate, within the experimental setting and results presented in the paper, and equally well-situated with the related works. The proposed approach is build on, particularly [R1] (and [R2]), both cited in the paper.

[R1] Namhoon Lee, Thalaiyasingam Ajanthan, and Philip HS Torr. Snip: Single-shot network pruning based on
connection sensitivity. arXiv preprint arXiv:1810.02340, 2018.

[R2] De Jorge P, Sanyal A, Behl HS, Torr PH, Rogez G, Dokania PK. Progressive skeletonization: Trimming more fat from a network at initialization. arXiv preprint arXiv:2006.09081. 2020 Jun 16.

**Requested Changes:**

* I think the authors should better justify the novelty of this work as compared to the related works.
* Please discusses, what are the considerations and implications of assumption made with a single global mask at initialization to align all updates in the same sparse subspace: "clients remain in a common subspace throughout training".
* The claim regarding the use of only a single minibatch per client should be justified and validated for performance guarantees.
* What would be the computational cost and complexity of the proposed approach, as compared with the related works. Can the framework handle system-level heterogeneity, i.e., difference in compute abilities of the clients?
* Discuss implications for using static mask and how to handle changing data distributions. I believe experimenting on this setting would bring both novelty and contributions to the work.
* Fig. 3 (b) is missing other baselines for fair assessment.

---

> ### Author Response · Authors · 2025-11-26
> **Response to Reviewer pA7V  (1/3)**
>
> We sincerely thank the Reviewer for their detailed review. We are encouraged that you find SSFL to be well-motivated, easy to follow and that you find the topic to be of significant interest.  We also thank the reviewer for acknowledging that our claims are accurate, convincing and supported by clear evidence and that our work demonstrates impressive performance gains. We address all of your concerns and suggestions below, and also update our manuscript (uploaded in the portal, all changes marked with red ink).
>
> ---
> ### **On novelty and contributions**
> > I think the authors should better justify the novelty of this work as compared to the related works...
>
> **We provide two different justifications on novelty** in regards to the literature, the field of Federated Learning (FL) and in light of TMLR's acceptance guidelines.
>
> **1. Our work tackles a novel and open question in the non-IID FL setting:**
>
> We agree our work builds on progress made by [1, 2, 3] in the specific case of centralized supervised learning setting, which we cite throughout the paper. ***However, we address fundamentally different and open problem in FL: whether pruning at initialization in a non-IID FL setting is possible*** without auxiliary datasets that breaks privacy, iterative multi-stage pruning, or additional hyperparameters, and if so, how to achieve it. **As Reviewer-YzoX notes, in this work we demonstrate this is indeed possible** and we explain our method and report our findings in this work.
>
> A couple of the core scientific questions we address are fundamentally federated in nature:
> - **How should saliency scores computed on heterogeneous, non-i.i.d. client data be aggregated** to identify a globally useful sparse subnetwork? SNIP [1] operates on centralized data; we demonstrate that data-weighted aggregation of local saliency scores (Eq. 3) enables effective one-shot mask discovery across heterogeneous non-IID clients.
> - **Can a global mask discovered at initialization effectively train across diverse client distributions?** Prior FL works [5,6] shows centralized methods often fail under non-i.i.d. conditions, or with the constraint of privacy preservation. Our experiments (Tables 1-3, Fig. 3, 4, 5) demonstrate that our masking strategy generalizes well even under extreme heterogeneity.
>
> Beyond these two, we also demonstrate the performance capabilities and advantages of using a stable global mask and learning in a common coordinate subspace.
>
> **Closing Notes on novelty**
> As a closing note, we reflect that SGD existed long before federated learning, yet FedAvg (McMahan et al., 2017) was a landmark contribution by demonstrating that SGD could be effectively adapted to federated settings with appropriate modifications (client sampling, weighted averaging, etc.). We believe similarly, demonstrating that gradient-based saliency can be effectively adapted to federated settings and empirical validation under non-i.i.d. conditions represents a meaningful contribution to the FL community.
>
>
> **2. Alignment with TMLR Acceptance Criteria and novelty**
>
> **While we strongly believe our work tackles an open problem in non-IID FL and makes novel contributions**, we also note that [TMLR's acceptance criteria](https://jmlr.org/tmlr/acceptance-criteria.html) emphasize  alignment of claims with evidence and potential audience interest as the primary basis for acceptance. The reviewer confirms our work meets both criteria: (1) claims are supported by convincing evidence, and (2) findings are of interest to TMLR's audience. We are, however, still committed to address any gaps if the reviewer still feels so.

---

> ### Author Response · Authors · 2025-11-26
> **Response to Reviewer pA7V (2/3)**
>
> &nbsp;
> ### **On the single global mask and common subspace assumption:**
> > - Please discuss, what are the considerations and implications of assumption made with a single global mask at initialization to align all updates in the same sparse subspace:
>
> The decision to use a single global mask at initialization is central to SSFL. It ensures all clients train in the same sparse subspace $\mathcal{S} = \{ v \in \mathbb{R}^d \mid v_j = 0 \text{ whenever } M_j = 0 \}$, where $M$ is the mask obtained from aggregated saliency scores. Our justification rests on three points:
> 1. **Preventing subspace mismatch.** If clients use different masks, their updates no longer live in the same parameter space. Client A may update a coordinate that Client B has pruned. When the server averages such incompatible updates, the result is unstable and often regrows density. This "incoherence" could make it difficult for the global model to preserve useful client-specific information.
> 2. **Experiment Isolates the value of a shared subspace.** To separate the effect of sharing the same subspace from the effect of saliency, Fig. 3(a) compares independent random masks vs. a global shared random mask. Even with no saliency at all, the shared mask clearly outperforms *client specific independent masks* (underlying mechanism of dynamic methods), which even collapses at higher sparsities. This shows that simply keeping clients aligned on which coordinates are active is could be a major factor.
>
> **Clarification on the number of minibatches used: We use $K$ minibatches**
> > *The claim regarding the use of only a single minibatch per client should be justified and validated for performance guarantees.*
>
> Thank you for raising this point. To clarify, while each client uses one local minibatch for efficiency, SSFL does **not** rely on a single minibatch overall. The global mask is computed by aggregating saliency scores from all $K=100$ clients, meaning it is estimated from **100 distinct minibatches** (about 1,600 samples on CIFAR-10).
>
> We acknowledge that this point was not sufficiently explained in the paper, and in response to the reviewer’s feedback we now explain this in the updated manuscript and perform a new **Oracle Mask Convergence experiments** (Sec. 4.2.3, Fig. 4(a-b)). We construct an oracle mask using the full dataset and measure how well masks estimated from $k$ minibatches match it. The mask error drops quickly and **plateaus around $k \approx 80$--$100$** on both CIFAR-10 and CIFAR-100, with little improvement beyond this point. Since SSFL aggregates exactly **$100$** minibatches (one per client), it operates precisely in this convergence regime. We further discuss how to adapt for lower or changing client numbers in FL setup in that section.
>
>
> ### **Computational Cost**
> >  What would be the computational cost and complexity of the proposed approach, as compared with the related works. Can the framework handle system-level heterogeneity, i.e., difference in compute abilities of the clients?
>
> We thank the reviewer for this recommendation and have added a rigorous complexity analysis in Appendix A.3. Unlike dynamic methods that incur high per-round overhead (SparsyFed: $\approx O(EB \cdot P)$ from weight re-parameterization at every forward pass, where $E$ is local epochs, $B$ is batches, and $P$ is parameters; DisPFL: $\approx O(P \log P)$ from prune-regrow cycles) or methods requiring costly training warm-ups (e.g., FLASH-SPDST with $E_d$ epochs), SSFL identifies a global mask with a one-time cost. This guarantees zero dynamic overhead during training, enabling hardware-accelerated sparse execution ($O(s \cdot \text{FLOPs})$ at density $s$) and symmetric communication savings ($O(sP)$) without the control-flow bottlenecks inherent to dynamic rewiring.
>
> > ...Can the framework handle system-level heterogeneity, i.e., difference in compute abilities of the clients?
>
> **System-Level Heterogeneity.** Regarding system heterogeneity, SSFL has the benefit of relying on hardware-friendly static masking as detailed in Appendix A.5 in the updated manuscript. Use of static masks (or infrequently changing masks, which SSFL can support as evident in our OOD adaptation) enables the use of hardware-optimized sparse kernels (e.g., cuSPARSELt) without constant recompiling (see A.5), allowing resource-constrained clients to participate that otherwise could not train dense or dynamically sparse models.
>
> Additionally, since SSFL shares global saliency scores rather than just masks, clients with different compute capabilities could select different sparsity levels from the same ranking (e.g., a resource-constrained client selects top-5% while a powerful client selects top-20%), as any top-$k$ mask is a superset of top-$l$ when $k > l$. This enables heterogeneous participation without sacrificing the benefits of a shared sparse subspace. However, such extensions and analysis are outside the scope for our current study, although logically feasible.

---

> ### Author Response · Authors · 2025-11-26
> **Response to Reviewer pA7V (3/3)**
>
> &nbsp;
> ### **Changing Distributions**
> > Discuss implications for using static mask and how to handle changing data distributions. I believe experimenting on this setting would bring both novelty and contributions to the work.
>
> We appreciate the suggestion regarding changing distributions. Since SSFL’s mask discovery is computationally inexpensive, adaptation is efficiently achieved via a "mask refresh" triggered by distribution shifts, avoiding the overhead of continuous updates.
>
> **We add new section (4.3) in main paper with experiments showcasing adaptation of the global model to OOD clients (new classes, previously unseen by global model)** introduced to the FL training at round $R=225$. We also formalize our strategy in newly added Appendix 6. and provide preliminary findings and leave detailed analysis for future work. Additionally, we note that while dynamic baselines (DisPFL, SparsyFed) theoretically handle OOD drift, their original studies do not conduct such experiments and rely solely on static benchmarks. We thank the reviewer for their review and for helping improve our work.
>
>
> #### References
> [1] Lee et al., "SNIP: Single-shot Network Pruning based on Connection Sensitivity," ICLR 2019.
>
> [2] Wang et al., "Picking Winning Tickets Before Training by Preserving Gradient Flow," ICLR 2020.
>
> [3] De Jorge et al., "Progressive Skeletonization: Trimming More Fat from a Network at Initialization," ICLR 2021.
>
> [4] Frankle & Carbin, "The Lottery Ticket Hypothesis: Finding Sparse, Trainable Neural Networks," ICLR 2019.
>
> [5] Distributed Pruning Towards Tiny Neural Networks in Federated Learning, 2023.
>
> [6] Model Pruning Enables Efficient Federated Learning on Edge Devices, 2022.

---

### Author Response · Authors · 2025-11-26
**Summary of Changes in updated manuscript during author-reviewer discussion phase**

We sincerely thank all the reviewers for their review and helping us make the manuscript better. We provide a summary of the updates we made to the manuscript taking into account the reviewer's comments, suggestions and questions.

**We make the following general changes:**
1. We organize the experimental section for better flow and coherence, and update and include new sections/explanations to help clarify aspects of the work based on the reviews, which we appreciate a lot.

**We conduct the following new experiments:**
1. **Mask convergence analysis** in section 4.2.3 of the main paper demonstrating the number of data samples needed to approximate the oracle mask (with 5 seeds). We show that our experiments leveraging $K=100$, falls in the convergent zone.
2. **Experiments on Larger Model:** We conduct experiments on the larger ResNet-50 Model and compare to best performing baseline model DisPFL and FedAvg (dense) baseline.
3. **Out of Distribution (OOD) Adaptation Extension and Experiments:** We add new results on OOD adaptation in Section 4.3 and include further details with a SSFL-OOD Algorithm-2  in Appendix A.6, as proof of concept for OOD adaptation.
4. We conduct new experiments to **study the effects of using Warm Up in Appendix A.7.**

**We further add the following sections discussing**
- Computation Analysis Section in Appendix A.3
- Explaining the benefits of stable static masks for Hardware acceleration in Appendix A.5 citing relevant literature and sources, which brings the potential for practical benefits like mask offline compression of masks, stored in hardware-optimized formats, and exploited by specialized kernels on GPU at resource constrained local clients, which is not possible or non-trivial for dynamic methods.
- Explaining our initialization schemes in appendix A.8.

**Open Source Code:** We shared our code as supplementary materials and if accepted, we are commited to publishing reproducible public codebase for the paper.

On top of the above changes, we believe we address all of the queries and concerns of the reviewers and perform new experiments whenever requested. We provide an updated manuscript marking all changes in red color. We thank the reviewers for their reivew and remain committed to address further queries. Thank you!

---

### Decision · Action_Editor_4QKP · 2025-12-11

**Recommendation:** Accept as is

**Audience:**

Yes

**Audience Explanation:**

Sparse subnetwork identification is an interesting problem, which is highly relevant to address on-device computation and communication payload, an important and interesting topic for TMLR community.

**Claims And Evidence:**

Yes

**Claims Explanation:**

In this paper, the authors have studied sparse subnetwork identification problem in a federated setting. They propose Salient Sparse Federated Learning  to obtain a globally shared sparse subnetwork at initialization using only local client data and develop a communication-efficient scheme to compute importance scores through aggregating local saliency scores in a single round of coordination.

After the rebuttal, two reviewers found the response and revised version satisfactory. Reviewer yNpy noted that "The mask selection based on the gradient at the initial point is stochastic and using such mask within training is problematic". I noted that the authors addressed the issue during rebuttal, clarified that "each client uses one minibatch for efficiency", and performed an experiment with an additional warm-up phase to show that there was no particular difference between warm-up and no warm-up.

Overall, I think the authors have sufficiently addressed the major concerns and recommend the paper be accepted as is.